# Precise in vivo functional analysis of DNA variants with base editing using ACEofBASEs target prediction

Alex Cornean[1,2†], Jakob Gierten[1,3,4†], Bettina Welz[1,2,4†], Juan Luis Mateo[5], Thomas Thumberger[1], Joachim Wittbrodt[1,4]*

[1]Centre for Organismal Studies, Heidelberg University, Heidelberg, Germany; [2]Heidelberg Biosciences International Graduate School (HBIGS), Heidelberg, Germany; [3]Department of Pediatric Cardiology, University Hospital Heidelberg, Heidelberg, Germany; [4]DZHK (German Centre for Cardiovascular Research), Heidelberg, Germany; [5]Deparment of Computer Science, University of Oviedo, Oviedo, Spain

*For correspondence:
jochen.wittbrodt@cos.uni-heidelberg.de

[†]These authors contributed equally to this work

Competing interest: The authors declare that no competing interests exist.

**Abstract** Single nucleotide variants (SNVs) are prevalent genetic factors shaping individual trait profiles and disease susceptibility. The recent development and optimizations of base editors, rubber and pencil genome editing tools now promise to enable direct functional assessment of SNVs in model organisms. However, the lack of bioinformatic tools aiding target prediction limits the application of base editing in vivo. Here, we provide a framework for adenine and cytosine base editing in medaka (*Oryzias latipes*) and zebrafish (*Danio rerio*), ideal for scalable validation studies. We developed an online base editing tool ACEofBASEs (a careful evaluation of base-edits), to facilitate decision-making by streamlining sgRNA design and performing off-target evaluation. We used state-of-the-art adenine (ABE) and cytosine base editors (CBE) in medaka and zebrafish to edit eye pigmentation genes and transgenic GFP function with high efficiencies. Base editing in the genes encoding troponin T and the potassium channel ERG faithfully recreated known cardiac phenotypes. Deep-sequencing of alleles revealed the abundance of intended edits in comparison to low levels of insertion or deletion (indel) events for ABE8e and evoBE4max. We finally validated missense mutations in novel candidate genes of congenital heart disease (CHD) *dapk3*, *ube2b*, *usp44*, and *ptpn11* in F0 and F1 for a subset of these target genes with genotype-phenotype correlation. This base editing framework applies to a wide range of SNV-susceptible traits accessible in fish, facilitating straight-forward candidate validation and prioritization for detailed mechanistic downstream studies.

## Editor's evaluation

This is an outstanding new method using base editor technology for introducing precise mutations in zebrafish and medaka vertebrate model systems. The approach is some of the strongest evidence to date that F0 functional analyses are becoming practical for screening work, with germline confirmation now rapidly possible as well due to the precise nature of the mutagenesis tool.

## Introduction

Single nucleotide variants (SNVs) are the most prevalent disease-causing alterations in human genes (*Claussnitzer et al., 2020*). While genomic studies continue to unravel SNVs rapidly, functional validation to rank their phenotypic impact in vivo remains the central bottleneck, hindering causal assessment. Precise genome editing at the single nucleotide level is a prerequisite to determine the

**eLife digest** DNA contains sequences of four different molecules known as bases that represent our genetic code. In a mutation called a single nucleotide variant (or SNV for short), a single base in the sequence is swapped for another base. This can lead the individual carrying this SNV to produce a slightly different version of a protein to that found in other people. This slightly different protein may not work properly, or may perform a different task. In recent years, researchers have identified thousands of SNVs in humans linked with congenital heart diseases, but the roles of many of these SNVs remain unclear.

Tools known as base editors allow researchers to efficiently modify single bases in DNA. Base editors use molecules known as short guide RNAs (or sgRNAs for short) to direct enzymes to specific positions in the DNA to swap, delete or insert a base. The sgRNAs need to be carefully designed to target the correct bases, however, which is a time consuming process. Furthermore, base editors were developed in cells grown in laboratories and so far only a few studies have demonstrated how they could be used in living animals.

To overcome these limitations, Cornean, Gierten, Welz et al. developed a framework for base editing in two species of fish that are often used as models in research, namely medaka and zebrafish. The framework uses existing base editors that swap individual target bases and a new online tool – referred to as ACEofBASEs – to help design the required sgRNAs. The team were able to use the framework to characterize the medaka equivalents of four SNVs that have been previously associated with congenital heart disease in humans.

The new framework developed here will help researchers to investigate the roles of SNVs in fish and other animals and validate human disease candidates. This approach could also be used to study the various ways that cells modify proteins by changing the specific bases involved in such modifications.

functional relevance and pathogenic contribution of SNVs in cell-based or animal disease models. Although CRISPR-Cas9 gene deletion is powerful in generating valuable insight into null phenotypes, a means to go beyond gene-level analysis to address the functional impact of single nucleotide changes directly is highly desirable.

Base editing has recently emerged as a single-nucleotide-level rubber and pencil tool (*Ravindran, 2019*), utilizing targeted hydrolytic deamination while largely omitting double-strand breaks (DSBs) (*Gaudelli et al., 2017*; *Komor et al., 2016*). Engineered proteins combining the PAM-specific action of a Cas9-D10A nickase or a dead Cas9 nuclease with cytidine or deoxyadenosine deaminases constitute the base editor complex. The deaminases thereby enable cytosine to thymine (C-to-T) or adenine to guanine (A-to-G) conversion within a predefined window of activity on the DNA segment (*Figure 1—figure supplement 1*), with limited indel formation. Established in human tissue culture, cytosine base editors (CBEs) and adenine base editors (ABEs) combined enable all transition mutations (*Gaudelli et al., 2017*; *Komor et al., 2016*). External development, rapid growth, and transparency of medaka (*Oryzias latipes*) and zebrafish (*Danio rerio*) embryos make these established vertebrate human disease models ideal for studying genetic variants' consequences in vivo (*Gut et al., 2017*; *Hammouda et al., 2021*; *Meyer et al., 2020*). Both ABEs and CBEs have been employed in zebrafish (*Carrington et al., 2020*; *Qin et al., 2018*; *Rosello et al., 2021b*; *Zhang et al., 2017*; *Zhao et al., 2020*), reaching up to 40% and 91% editing efficiencies, respectively. Notably, *Rosello et al., 2021b* have recently established a proof-of-concept for cytosine base editing to study developmental and human disease variants in fish. CRISPR/Cas9-mediated gene knockout can be readily used to discern gene function in fish in F0 enabled by high Cas9 nuclease efficiencies and robust workflows that include versatile sgRNA design tools (*Hoshijima et al., 2019*; *Kroll et al., 2021*; *Stemmer et al., 2015*). Various iterations of improvements yielding highly efficient base editing tools in vitro, also led to high C-to-T editing efficiencies in some of the loci tested (*Rosello et al., 2021b*). However, for both CBE and ABE systems tested in fish, reliable information for base editor selection to consistently achieve high efficiencies is absent, impeding the validation of specific DNA variants by linking these to specific traits in F0. Moreover, the lack of user-friendly web-based software for sgRNA design and selection for target DNA variants has hampered routine applications of base editing in fish.

Here, we present a comprehensive workflow for adenine and cytosine base editing in medaka and zebrafish. To design and evaluate nucleotide and codon changes and predict potential off-target sites, we built on our CRISPR-design tool CCTop (*Stemmer et al., 2015*), now including a careful evaluation of base-edits (ACEofBASEs). We employ this simple, efficient, and easy-to-use software to design sgRNAs for the state-of-the-art cytosine base editors BE4-Gam (*Komor et al., 2017*), ancBE4max (*Koblan et al., 2018*), and evoBE4max (evoAPOBEC1-BE4max; *Thuronyi et al., 2019*), as well as the adenine base editor ABE8e (*Richter et al., 2020*). Editing efficiencies close to homozygosity in injected (F0) embryos allowed us to directly link specific missense mutations or premature termination codons (PTC) to phenotypic outcomes, with a recapitulation of expected phenotypes. To address the relative distribution of edited alleles we performed Amplicon deep-sequencing (Amplicon-seq). Editing of the eye pigmentation gene *oca2* with cytosine or adenine base editors in F0 yielded high on-target editing (79%) and editor dependent low to moderate levels of indels (8–14%). Notably, introducing a single missense mutation in an ultra-conserved pore-domain of *kcnh6a* (*Ol* ERG), an essential potassium channel gene, resulted in a non-contractile ventricle with striking secondary morphological defects. We further used base editing in medaka to address SNVs associated with human congenital heart disease (CHD). The efficient introduction of conserved missense mutations in four novel candidate genes, *dapk3*, *ube2b*, *usp44*, and *ptpn11* in F0, and the analysis of germline-transmitted alleles in F1 for a subset of genes resulted in comparable phenotypic consequences, uncovering a functional role of these genes in cardiovascular development.

The conservation of a significant proportion of coding SNVs in fish enables rapidly prioritizing disease variants in F0 and validating their initially observed functional relevance in F1. The presented software and workflow for base editing in medaka and zebrafish coupled to our experimental data allow rational editor choice for in vivo SNV validation experiments. The efficiencies demonstrated are compatible with F0 phenotype quantification, a reliable and robust shortcut to endpoints of detailed developmental studies, and rapid candidate validation in human genetics.

## Results

### ACEofBASEs for base editor sgRNA design

The lack of comprehensive software tools for base editor sgRNA design and on- and off-target editing predictions significantly limits widespread base editing applications. We developed the online software tool ACEofBASEs as an extension of our CRISPR-Cas prediction tool CCTop (*Stemmer et al., 2015*) for straightforward sgRNA design for CBEs and ABEs. ACEofBASEs identifies all sgRNA target sites of a given query nucleotide sequence with adenine or cytosine residues present in the respective base-editing windows. A dropdown menu enables the selection of one of four recent state-of-the-art base editors: BE4-Gam, ancBE4max, evoBE4max, and ABE8e (*Figure 1a*). The standard base editing window parameters are set according to in vitro data, which differentiates between a window with high (blue) and observed activity (light blue) (*Arbab et al., 2020*; *Huang et al., 2021*; *Koblan et al., 2018*; *Komor et al., 2017*; *Richter et al., 2020*; *Thuronyi et al., 2019*, *Figure 1—figure supplement 2*). All detected sgRNAs for a particular target site are ranked considering the dinucleotide context and the editing window (displayed on a yellow-orange-brown scale; *Figure 1—figure supplement 2*). Alternatively, the limits of the base editing window can be freely adjusted (without sgRNA ranking, *Figure 1a*). The specifications of the oligonucleotides for sgRNA cloning are customizable in the user interface. Off-target prediction in the selected target genome follows the same routine used in CCTop (*Stemmer et al., 2015*) but only presents sgRNA off-target sites with adenines or cytosines present in the respective base editing window.

ACEofBASEs expects the query sequence to be coding and assumes that the ORF starts at the first nucleotide provided and can, if required, also deal with intronic sequences. The reading frame can be adjusted in the dropdown menu and the codons, translation and predicted sequence alterations are highlighted on the results page. The sgRNA display includes base edited amino acid sequence changes specified as missense (purple) or nonsense (black) mutations for the selected ORF (*Figure 1b*). Avoiding off-target editing is essential for organismal F0 screens as well as in cell-based assays, and consequently, the efficient off-target prediction represents a crucial feature of any base editing software tool. The ACEofBASEs overview of sgRNA targets provides this information at a

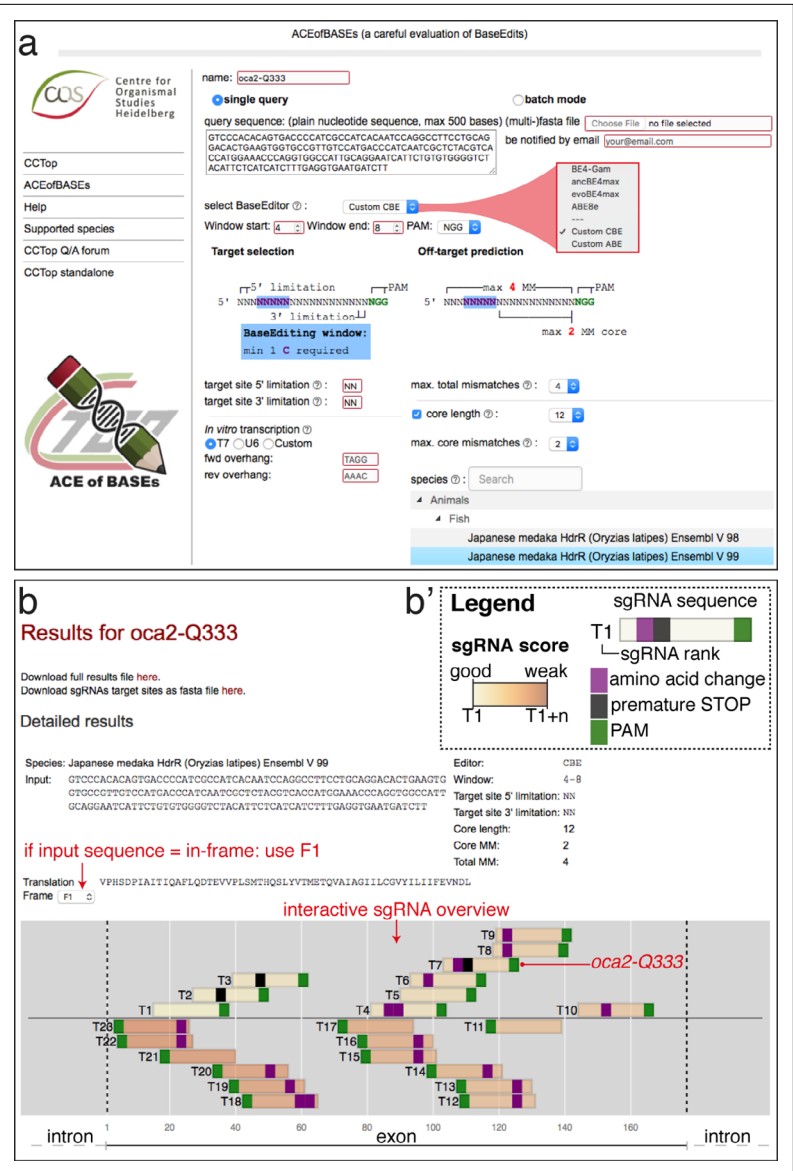

**Figure 1.** ACEofBASEs (a careful evaluation of base edits) enables simple and tailored use of base editors. (**a**) User interface of the ACEofBASEs base editing design tool with base editor choice dropdown menu and a compendium of model species to select from. (**b**) Results page of the ACEofBASEs design tool for cytosine base editing of the *Oryzias latipes oca2* locus. Using an in-frame sequence of the target site will directly provide the translation frame F1. Alternatively, frames can be selected from the dropdown menu. All sgRNA target sites found in the query are shown with potential amino acid change (magenta box) or nonsense mutation (PTCs, black box); here: standard editing window: nucleotides 4–8 on the protospacer; PAM: positions 21–23. For off-target prediction, a comprehensive list of potential off-target sites that contain an A or C in the respective base editing window is provided per sgRNA target. Potential off-target sites are sorted according to a position-weighted likelihood to introduce an off-target, that is the closer a mismatch at the potential off-target site to the PAM, the more unlikely this site is falsely edited (*Stemmer et al., 2015*).

The online version of this article includes the following figure supplement(s) for figure 1:

**Figure supplement 1.** Basic mechanism of cytosine base editing exemplifying the base editing principle.

**Figure supplement 2.** sgRNA score explanation and example.

**Figure supplement 3.** Details on selected sgRNA in ACEofBASEs.

glance and details each sgRNA with potential off-target coordinates and the type of genomic location (intergenic, intronic, or exonic) (*Figure 1—figure supplement 2*).

To address the power of ACEofBASEs, we used it to select sgRNAs creating easily scorable loss-of-function alleles by introducing defined PTCs or missense mutations by one of the four state-of-the-art adenine and cytosine base editors.

## Biallelic PTC and missense mutations through cytosine and adenine base editing in fish

Single nucleotide changes through PTCs resulting in nonsense-mediated decay or truncated proteins, and missense mutations substituting individual amino acids, can result in many of functionally relevant phenotypes. We investigated editing efficiencies for cytosine editors by their potential of introducing PTCs, and in contrast, for adenine base editors through the installation of missense mutations resulting in loss-of-function phenotypes. We compared the original BE4-Gam (*Komor et al., 2017*) to two next-generation CBEs, ancBE4max (*Koblan et al., 2018*) and evoBE4max (*Thuronyi et al., 2019*), the latter of which had not been previously tested in fish (*Figure 2a*). In addition, we also employed the highly processive adenine base editor ABE8e (*Richter et al., 2020*), potentially overcoming target constraints reported for ABE7.10 in zebrafish (*Qin et al., 2018*).

An acute and early onset of base editing activity and high base conversion rates are prerequisites for efficient biallelic editing causing reliable phenotypes in F0 experiments. We examined the in vivo efficiencies of state-of-the-art base editors by editing the *oculocutaneous albinism II* gene (*oca2*) required for eye pigmentation, a gene that had been successfully targeted with BE4-Gam in medaka (*Thumberger et al., 2022*). The loss of retinal pigmentation provides a scorable readout to address the efficacy of bi-allelic base editing or Cas9 based knock-out experiments (*Lischik et al., 2019*; *Zhang et al., 2017*), since a single allele of *oca2* is sufficient to generate normal retinal pigmentation and only cells affected in both alleles will exhibit fail to get pigmented. Within conserved regions of *oca2* (*Figure 2—figure supplement 1a*), ACEofBASEs proposed a sgRNA to introduce a PTC (*p.[T332I; Q333X]*) or two missense mutations (*p.[T332A; Q333R]*) in medaka with CBEs or ABEs, respectively.

We used microinjection of mRNA to deliver base editors and sgRNAs into one-cell stage medaka embryos. Phenotyping of base edited embryos, henceforth referred to as 'editants', at 4 dpf revealed a spectrum from wild-type to complete loss of ocular pigmentation divided into five arbitrary categories (*Figure 2b–c*). While BE4-Gam produced at least one pigment-free clone in almost all editants, overall, pigmentation was still prevalent. In sharp contrast, both ancBE4max and evoBE4max in a large fraction of editants led to the almost complete eye pigmentation loss (*Figure 2d*). Sanger-sequencing of randomly pooled editants reflected the same trend, that is, substantially remaining wild-type C peaks (protospacer position 5–7) for BE4-Gam, but complete C-to-T conversion achieved at protospacer position 6–7 using ancBE4max and evoBE4max (*Figure 2e*). Sanger sequencing did not reveal other DNA sequence changes such as indels or unwanted editing around the locus (*Figure 2—figure supplement 1b*). Quantification of Sanger sequencing results by editR (*Kluesner et al., 2018*) indicated editing efficiencies at C7 (c.997C > T, i.e., p.Q333X) of 29.3% ± 7.4% (n = 3), 93.8% ± 7.9% (n = 5), and 93.3% ± 9.8% (n = 3) for BE4-Gam, ancBE4max and evoBE4max, respectively (*Figure 2f*, *Supplementary file 1*). Interestingly, evoBE4max showed reduced C-to-T conversion at C5 within an AC dinucleotide context (*Figure 2e–f*), in line with its context-dependent substrate C editing efficiency detailed in-depth in cell culture (*Arbab et al., 2020*). Accordingly, we observed a reduced average conversion rate of 20.7% ± 0.6% at target C5 in an AC context for evoBE4max, compared to 29.3% ± 6.7% and 86.4% ± 11.5%, for BE4-Gam and ancBE4max, respectively (*Figure 2f*). In addition to ancBE4max and evoBE4max displaying prominently increased editing activity, a slight increase in aberrant F0 phenotypes was observed with both editors (*Figure 2—figure supplement 1c*).

In summary, ancBE4max and evoBE4max are highly efficient in deaminating targeted cytosines in vivo, demonstrated by introducing a PTC in *oca2*, resulting in near-complete loss of eye pigmentation in medaka embryos compatible with accelerated F0 genotype-phenotype studies.

We tested the effects of single (*oca2-Q333*) and pooled sgRNA (*oca2-Q256, T306, Q333*) editing events, aided by ACEofBASEs, with ABE8e to compare the impact of multiple vs a single amino acid exchange. Following our described workflow and categorization with ABE8e, we noted that pooled *oca2*-sgRNA injections led to a higher fraction (32 of 32) of embryos with reduced eye pigmentation, in contrast to the oca2-Q333 injection alone (14 of 48) (*Figure 2d*). The surge in expected

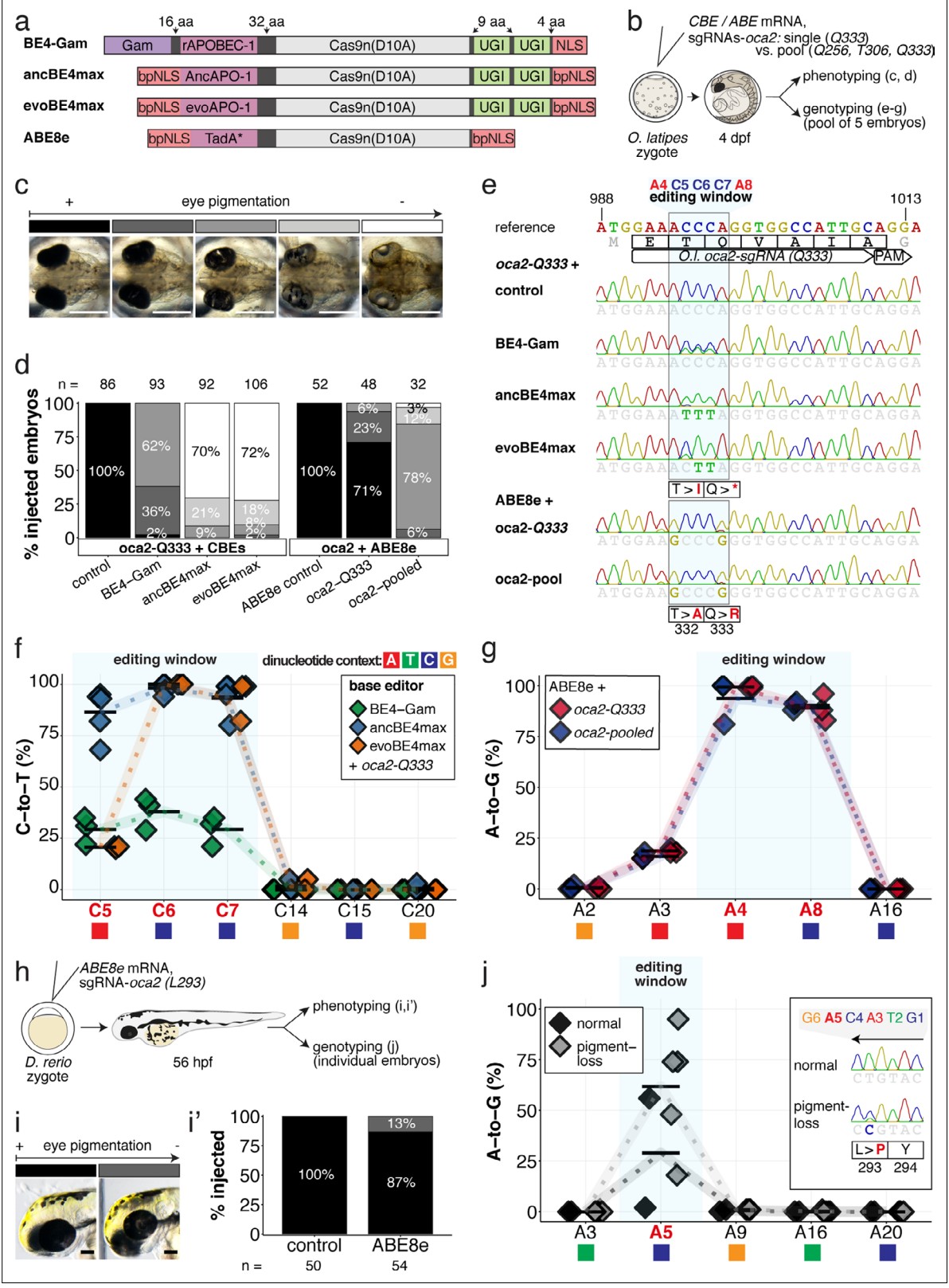

**Figure 2.** Somatic cytosine and adenine base editing at the *oculocutaneous albinism II (oca2)* locus in medaka and zebrafish allows direct functional assessment. (**a**) Schematic diagram of the cytosine base editors BE4-Gam, ancBE4max and evoBE4max (evoAPOBEC1-BE4max) and the adenine base editor ABE8e. Cas9n-D10A nickase (light grey) with N-terminally linked cytidine or deoxyadenosine deaminase (pink) and C-terminal SV40 or bipartite (bp) nuclear localization sequence (NLS, red). All except BE4-Gam also contain the bpNLS N-terminally. CBEs contain variations of the rat APOBEC-1

*Figure 2 continued on next page*

*Figure 2 continued*

cytidine deaminase, whereas ABE8e contains the TadA* domain (tRNA adenine deaminase), CBEs further contain C-terminally linked Uracil glycosylase inhibitors (UGI, green). Gam protein from bacteriophage *Mu* (purple) and linkers of varying lengths (dark grey). (**b**) Scheme of the experimental workflow. Cytosine or adenine base editor (CBE/ABE) mRNA and *oca2-Q333* or a pool of three *oca2-sgRNAs* (–Q256, –T306, –Q333) were injected into the cell of a medaka zygote. Control injections only contained *oca2-Q333* or *ABE8e* mRNA. (**c**) Phenotypic inspection of eye pigmentation was performed at 4 dpf (dorsal view). (**d**) Grouped and quantified pigmentation phenotypes shown for BE4-Gam, ancBE4max, evoBE4max, and ABE8e experiments. Control only contains *oca2-Q333* sgRNA. n shown excludes embryos that are otherwise abnormal or dead, with abnormality rate given in supplement 1c. (**e**) Exemplary Sanger sequencing reads for each experimental condition, obtained from a pool of five randomly selected embryos at the *oca2-Q333* locus. (**f–g**) Quantification of Sanger sequencing reads (by EditR, *Kluesner et al., 2018*) for BE4-Gam (n = 3), ancBE4max (n = 5) and evoBE4max (n = 3) (**f**), and ABE8e for single (n = 3) and pooled oca2-sgRNA experiments (n = 3) (**g**). Pools of five embryos per data point summarizes editing efficiencies. Mean data points are summarized in *Supplementary files 1 and 2*. To highlight the dinucleotide context, the nucleotide preceding the target C or A is shown by red (**A**), green (**T**), blue (**C**) and yellow (**G**) squares below the respective C or A. (**h**) Microinjections into the yolk of one-cell stage zebrafish were performed with *ABE8e* mRNA and *oca2-L293* sgRNA. Zebrafish larvae were phenotypically analyzed at 56 hpf and individual larvae were subsequently genotyped. (**i-i'**) Larvae were scored as without ('normal', black) or with loss of eye pigment (grey). (**j**) Sanger sequencing on individually scored larvae was analyzed by EditR and plotted according to phenotype. Scale bars = 400 µm (**c**) or 100 µm (**i**). dpf / hpf = days/hours post fertilization.

The online version of this article includes the following figure supplement(s) for figure 2:

**Figure supplement 1.** Somatic cytosine and adenine base editing at the medaka *oculocutaneous albinism II* (*oca2*) gene occurs in the absence of detectable indels and DNA off-target editing.

**Figure supplement 2.** ABE8e efficiently introduces A-to-G mediated missense mutations in medaka and zebrafish.

phenotypes, however, coincided with an increase in aberrant phenotypes (*Figure 2—figure supplement 1c*). We observed close to homozygous editing resulting in missense mutations p.T332A (99.3% ± 0.6%, 93.7% ± 11.0%) and p.Q333R (89.0% ± 6.6%, 90.3% ± 2.1%) by single and pooled *oca2* experiments, respectively (*Figure 2e–g*, *Supplementary file 2*). For pooled sgRNA injections, we also observed efficient editing at oca2-Q256 (100%) and -T306 (52.3% ± 2.3%) (*Figure 2—figure supplement 2a-b*), which may implicate a causative role of oca2-Q256 in the increased phenotypic fraction. Closer analysis of *oca2-Q333* editants with evoBE4max and ABE8e revealed no off-target (OT) editing at three partially matching sites indicated by ACEofBASEs (*Figure 1—figure supplement 3*, *Figure 2—figure supplement 1d*).

The inherent limitations of Sanger sequencing did not allow to resolve both editing <5% or monoallelic edits (*Kluesner et al., 2018*). To overcome this limitation, we analyzed a subset of *oca2-Q333* individuals by Amplicon-seq (*Figure 3*). The number of overall modified alleles resembled the editing efficiencies initially determined by Sanger sequencing. The indel frequency is highly base editor and ranges from 2.1% (BE4-Gam), 8.0% ± 2.3% (ABE8e), 14.4% (evoBE4max) to 19.9% (ancBE4max) (*Figure 3a–b*, *Supplementary file 4*). At the *oca2-Q333* sgRNA target site deletions occurred predominantly around the Cas9 nickase nick site (protospacer positions 17/18) (*Figure 3c-d*, *Figure 3—figure supplements 1 and 2*). Taken together, our Amplicon-seq analysis of the different base editors at the *oca2* locus revealed high on-target editing efficiencies, with editor-specific low-to-moderate indel frequencies.

Next, we compared the performance of ABE8e in injected zebrafish embryos and determined the state of eye pigmentation of successfully gastrulating embryos at 56 hpf. Here, seven out of 54 editants (13%) displayed reduced eye pigmentation (*Figure 2h–i'*). Genotyping of individual embryos, grouped according to the pigmentation phenotype (normal vs. pigment-loss), revealed a direct correlation between phenotype and A-to-G transition efficiency with 29.0% ± 29.8% and 61.8% ± 29.6% at A5 (*Figure 2j*), with no aberrant sequence changes in the locus (*Figure 2—figure supplement 2c*). Moreover, we observed editing of up to 95% introducing a close to homozygous missense p.L293P mutation, indicating that, similarly to medaka *oca2-Q333* editing, these missense mutations do not fully impair protein function.

Taken together, we showed that ABE8e efficiently introduced missense mutations with all four sgRNAs tested in medaka and zebrafish. To examine whether missense mutations lead to fully penetrant phenotypes in F0, we next addressed the consequences of a specific missense loss-of-function mutation on transgenic GFP.

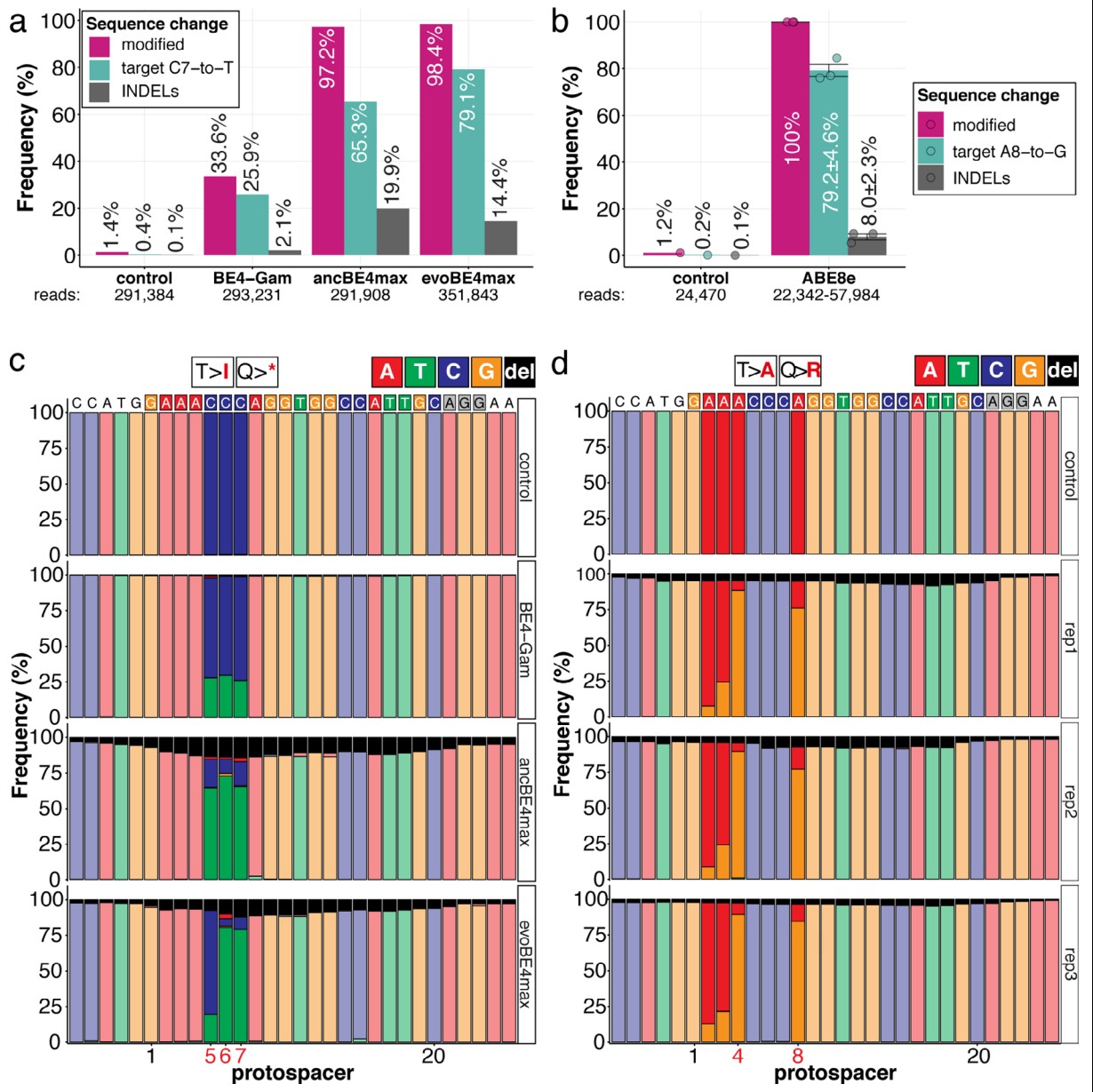

**Figure 3.** Amplicon-seq of cytosine base editors (**a, c**) and ABE8e (**b, d**) reveals prominent on-target editing efficiencies with low- to moderate levels of indels. Genomic DNA samples from *oca2-Q333* base editing experiments (*Figure 2b–g*) were used to query the outcome of intended base editing and indel formation by quantitative means by Illumina sequencing of the target region. Note: for oca2-Q333 control, BE4-Gam, ancBE4max, and evoBE4max, two pools of five embryos were used as sample input (**a, c**), whereas all three biological replicates of ABE8e samples were sequenced separately (**b, d**). The proportion of all reads aligned per sample to a reference is plotted, distinguishing (1) all modified reads, (2) target cytosine (**C7, a**) or adenine (**A8, b**) nucleotide changes and (3) INDELs. The number of reads shown (**a, b**) refers to all aligned Illumina reads per sample. The frequencies of base calls at the *oca2-Q333* sgRNA target site ± 5 bp is shown for the three different cytosine editors (**c**) and ABE8e (**d**).

The online version of this article includes the following figure supplement(s) for figure 3:

**Figure supplement 1.** Allele frequency of CBE experiments at the *oca2-Q333* locus.

**Figure supplement 2.** Allele frequency of ABE8e experiments at the *oca2-Q333* locus.

## The highly processive ABE8e efficiently introduces missense mutations inactivating GFP fluorescence

To expand our initial analysis on ABE8e base editing efficiency, we took advantage of the deep functional annotation of individual amino acids in GFP (*Fu et al., 2015*; *Li et al., 1997*; *Patterson et al.,*

*1997*) as target protein.

We assayed the loss of GFP fluorescence in a medaka EGFP and mCherry double transgenic line with heart-specific fluorophore expression (*myl7::EGFP, myl7::H2A-mCherry*) (*Hammouda et al., 2021*). Co-injection of ABE8e mRNA with the *GFP-C71* sgRNA into one-cell stage embryos resulted in the complete loss of GFP fluorescence in all hearts assayed at 4 dpf (n = 41), indicating quantitative editing as also reflected by the absence of mosaicism.

Cells in the medaka four-cell embryo form a syncytium allowing limited diffusion of the base editing machinery provided to one of the blastomeres. Only when forced by injection into a single blastomere at the four-cell stage, mosaicism, apparent by speckled GFP-positive hearts, could be forced in all embryos analyzed (100%, n = 23) (*Figure 4a–d*), further underpinning the high editing efficiency.

Sanger sequencing and EditR quantification confirmed the strikingly high efficiencies in introducing the intended p.C71R missense mutation (97.0% ± 4.4% and 90.0% ± 7.8%), as well as a notable bystander edit causing p.V69A missense mutation (61.2 ± 9.4 and 12.6% ± 8.1%) for one-cell and four-cell stage injections, respectively (*Figure 4e–f*; *Supplementary file 2*). Amplicon-seq performed on individual embryos injected at the one-cell stage confirmed high conversion efficiencies of 85.9% ± 15.3% with 15.8% ± 17.5% indels (*Figure 4g*, *Figure 4—figure supplement 1*).

The highly efficient editing, even when only provided to one of the four cells, underscored the high in vivo potency of ABE8e even at reduced levels. The high, almost quantitative activity of both CBEs and ABE8e, even at low concentrations in vivo, led us to address phenotypic consequences of editing individual sites in established as well as so far, unknown targets.

## Functional intervention in F0 by CBE induced stop-gain mutations in cardiovascular troponin T

The high efficiency of base editing already in the injected generation opens the door for functional evaluation of candidates rapidly uncovered by extensive population-scale sequencing studies of cardiovascular disease (CVD) (*Ingles et al., 2020*; *Kathiresan and Srivastava, 2012*; *Pierpont et al., 2018*). The understanding of the mechanistic contribution of those variants holds a great promise for prevention and precision medicine.

We first validated the relevance of in vivo base editing in the injected generation by phenocopying reference heart phenotypes associated with fully penetrant, recessive lethal genes from fish (*Chen et al., 1996*; *Meyer et al., 2020*; *Stainier et al., 1996*). We targeted *troponin T type 2 a* (*tnnt2a*) that, when mutated, results in silent, non-contractile hearts in zebrafish *sih* mutants (*Sehnert et al., 2002*) as well as in medaka 'crispants' of *tnnt2a* (*Meyer et al., 2020*). We addressed the ACEofBASEs predicted conversion of Q114 into a PTC mediated using a previously employed sgRNA (*Figure 5a*). We co-injected this sgRNA with mRNAs encoding the different CBEs and analyzed the resulting phenotypes leading to the impairment of cardiac contractility at various degrees (*Figure 5b*, *Video 1*).

Depending on the base editor employed, the penetrance of the phenotype drastically increased, from BE4-Gam to ancBE4max and evoBE4max (*Figure 5c*), reflecting the different editing efficiencies (c.340C > T resulting in p.Q114X) of 0% (BE4-Gam), 27.8% ± 16.8% (ancBE4max), and 85.9% ± 23.5% (evoBE4max) (*Supplementary file 1*). Analysis of the editing efficiencies in the silent heart group indicated prevalent homozygous C8-to-T edits (p.Q114X) by evoBE4max (*Figure 5d–d'*). We next analyzed and confirmed the high efficiencies of evoBE4max experiments by Illumina sequencing in a subset of samples. We observed remarkable editing efficiencies of 89.3% ± 6.6% and low indel rates (7.7% ± 7.3%), underscoring the potency of evoBE4max in this setting (*Figure 5c*, *Figure 5—figure supplement 1*).

The observed medaka 'silent heart' phenotypes, affecting both, overall development and cardiac contractility, were distinct and differ from the phenotype previously described for zebrafish *sih* mutants, which are characterized by a specific loss of cardiac contractility (*Sehnert et al., 2002*). We therefore used base editing at additional sites to further corroborate the role of medaka *tnnt2a*. We created a PTC at an alternative position employing a sgRNA (*tnnt2a-W201*), co-injected with the evoBE4max editor. Forty of 44 (91%) of those embryos displayed the same medaka silent heart phenotype, independently confirming the results at the initial editing site (*tnnt2a-Q114*). Sanger sequencing confirmed the efficient installation of the intended PTC (72.0% ± 34.1%) at the *tnnt2a-W201 site* (c.C603 >T resulting in p.W201X) (*Figure 5—figure supplement 2*).

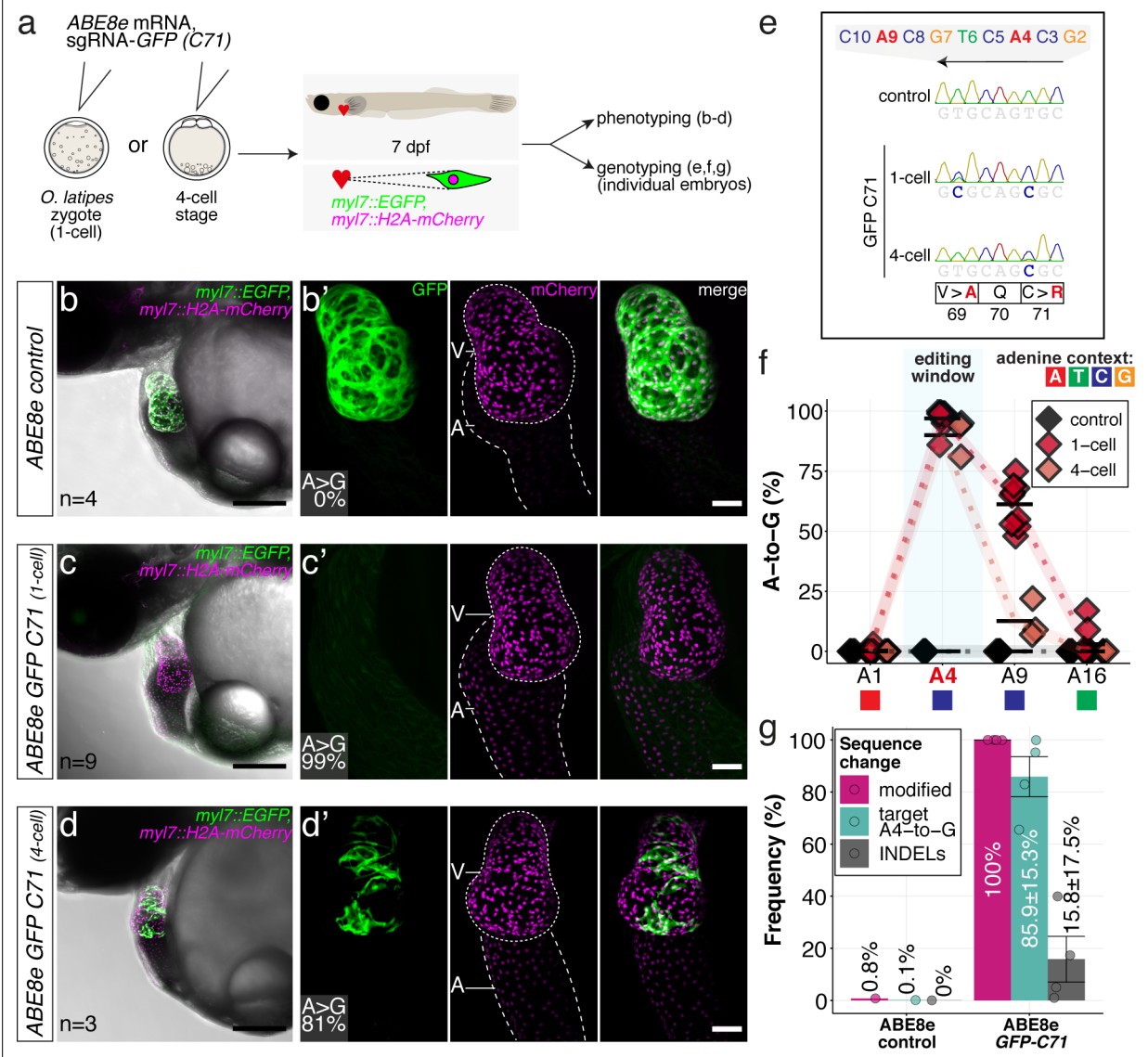

**Figure 4.** ABE8e efficiently introduces missense mutation and completely abolishes GFP fluorescence in F0. (**a**) Co-injection of *ABE8e* mRNA with *GFP-C71* sgRNA into a single cell of one-cell or four-cell stage medaka embryos (*myl7::EGFP, myl7::H2A-mCherry*). Control injections with ABE8e mRNA. Scoring of fluorescence was performed at 7 dpf followed by genotyping of each individual embryo. Confocal microscopy of chemically arrested hearts (representative images) at 7 dpf (lateral view with V = ventricle, A = atrium). Overview images, overlaid with transmitted light, show maximum z-projections of optical slices acquired with a z-step size of 5 μm. Scale bar = 200 μm (**b–d**). Close-up images show maximum z-projections of optical slices acquired with a z-step size of 1 μm. Note the display of A-to-G conversion rates for A4 causing the p.C71R missense mutation (see g) and *Supplementary file 2*. Scale bar = 50 μm (**b-d'**). (**e–f**) Quantification of Sanger sequencing reads show close to homozygosity rates of A-to-G transversions installing the C71R missense mutation (*Supplementary file 2*). Note: sgRNA *GFP-C71* targets the complementary strand (arrow in f). To highlight the dinucleotide context, the nucleotide preceding the target A is shown by red (**A**), green (**T**), blue (**C**) and yellow (**G**) squares below the respective A. dpf = days post fertilization. (**g**) Amplicon-seq of the target region a subset (n = 4) of 1 cell stage ABE8e experiment gDNA samples (single embryos) was used to quantify the outcome of intended base editing and indel formation. Aligned Illumina-reads analyzed, 14,653 (control); 23,201 (ABE8e rep1); 10,696 (ABE8e rep2); 66,311 (ABE8e rep3); 48,126 (ABE8e rep4).

The online version of this article includes the following figure supplement(s) for figure 4:

**Figure supplement 1.** Sequence composition following Amplicon-seq of ABE8e *GFP-C71* editants surrounding the *GFP-C71* sgRNA target site ±5 bp.

To address whether the F0 editant phenotype is a good proxy for the phenotype of a stable *tnnt2a* mutant, we crossed the *tnnt2a-Q114* allele to homozygosity. To establish heterozygous founders, we injected the *tnnt2a-Q114* sgRNA at reduced concentrations of evoBE4max mRNA (5–15 ng/μl) and raised the survivors of the injection to adulthood (*Figure 5f*). We crossed sequence-validated F0

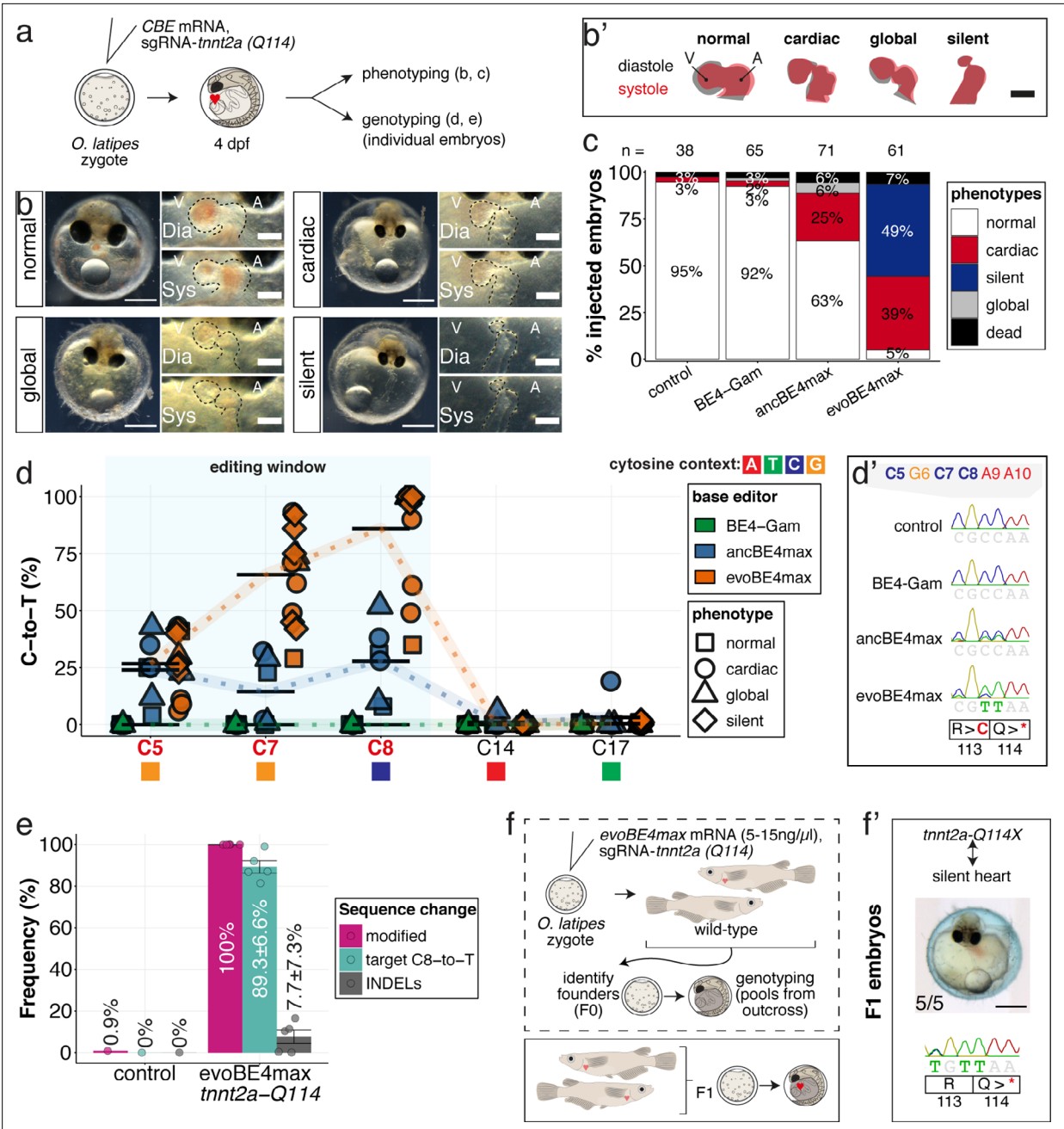

**Figure 5.** Stop-gain mutation in medaka troponin T gene by cytosine base editing accurately recreates mutant phenotypes in F0. (**a**) Schematic diagram of protocol after co-injection of cytosine editor mRNA (comparing BE4-Gam, ancBE4max, evoBE4max) with sgRNA *tnnt2a-Q114* (PTC) into one-cell stage medaka embryos; control injections only contained the sgRNA. (**b**) Editing of *tnnt2a* resulted in a range of phenotypes classified into five categories, including general morphogenic (global), dysmorphic but still functional heart chambers (cardiac), and non-contractile hearts (silent), where cardiac and silent phenotype groups displayed homogeneously additional developmental retardation. Scale bar = 400 µm (overview) and 100 µm (zoom-in). Ventricle (**V**), atrium (**A**), diastole (Dia) and systole (Sys) are indicated. (**b'**) Representative scheme of fractional shortening of the heart chambers in specified phenotype groups highlighting significant morphological consequences (small ventricle) in the silent heart group. (**c**) Fraction of phenotype scores as a consequence of cytosine base editor injections. (**d**) Summary of editor type-specific C-to-T conversion efficiencies relative to the target C protospacer position grouped by phenotype class for BE4-Gam (n = 6), ancBE4max (n = 6) and evoBE4max (n = 12). To highlight the dinucleotide context, the nucleotide preceding the target C is shown by red (**A**), green (**T**), blue (**C**), and yellow (**G**) squares below the respective C. (**d'**) Example Sanger sequencing reads of single edited embryos with resulting missense and stop-gain mutations through editing at C5, C7, and C8, respectively (**e**) Amplicon-seq of the target region of a subset (n = 5) of evoBE4max edited gDNA samples (single embryos) quantified target C8-to-T editing as well as indel frequencies. Aligned Illumina-reads analyzed, 7094 (control); 11,557 (evoBE4max rep1); 2561 (evoBE4max rep2);

*Figure 5 continued on next page*

*Figure 5 continued*

2481 (evoBE4max rep3); 37,751 (evoBE4max rep4); 48,791 (evoBE4max rep5). (**f**) Phenotypic analysis of F1 *tnnt2a-Q114X* mutants revealed complete penetrance (n = 5) of the silent heart phenotype with same phenotypic profile as for F0 edits. dpf = days post fertilization.

The online version of this article includes the following figure supplement(s) for figure 5:

**Figure supplement 1.** Sequence composition determined by Amplicon-seq of evoBE4max *tnnt2a-Q114* editants surrounding the *tnnt2a-Q114* sgRNA target site ± 5 bp.

**Figure supplement 2.** The installation of an additional PTC at *tnnt2-W201X* with evoBE4max leads to a recapitulation of the silent heart phenotype.

**Figure supplement 3.** Cytosine base editing enables efficient installation of PTCs or missense mutations in two additional cardiac genes.

founder fish with germline transmission of the Q114X allele and observed a silent heart phenotype in all homozygous tnnt2a-Q114X F1 offspring (n = 5) that matched the initially observed F0 phenotypes (*Figure 5f'*).

Genetic analysis revealed identical phenotypes when comparing the F1 homozygous mutants to the F0 editants. This underscores the high specificity of editing already in the injected generation. Our data also highlight an apparently different biological impact of the loss of *tnnt2a* in medaka when compared to zebrafish, likely due to the underlying evolutionary divergence between the two fish species.

In summary, highly efficient base editing of *tnnt2a* in injected F0 embryos allowed to establish and analyze a distinct phenotype in F0 editants that is fully resembled in the F1 homozygous mutants.

Expanding our proof of concept analyses, we determined editing efficiencies using a sgRNA for *tnnt2c* (paralog of silent heart-associated *tnnt2a*) and two sgRNAs for *s1pr2*, a sphingolipid receptor known to direct the midline migration of the bilateral heart precursors disrupted in the *miles apart* (*mil*) zebrafish mutant (*Kupperman et al., 2000*). We observed average C-to-T editing efficiencies of 38–88% across these loci in F0 (*Figure 5—figure supplement 3*, *Supplementary file 1*), further substantiating the high editing power in F0, allowing us to address the role of individual amino acids in complex proteins.

## In vivo modeling of heritable LQTS mutations using cytosine and adenine base editing

We next specifically targeted crucial amino acids in the cardiac-specific potassium channel ERG, essential for cardiac repolarization (*Arnaout et al., 2007*; *Hassel et al., 2008*). The ERG channel is a homo-tetramer with each subunit consisting of six alpha-helical transmembrane domains S1-S4 forming a voltage sensor and the pore-forming domains S5-S6 (*Figure 6—figure supplement 1*; *Wang and MacKinnon, 2017*; *Zhang et al., 2004*). Functional impairment of the human potassium channel *Hs* ERG (*KCNH2*) causes inherited or acquired long QT syndrome (LQTS), respectively, associated with sudden cardiac death (*Abriel and Zaklyazminskaya, 2013*).

Using CBEs, we revealed the loss-of-function phenotype in *kcnh6a* editants by PTC introduction in the F0 and F1 generation (*Figure 6—figure supplements 1–3*, *Video 2*). To specifically alter the voltage-sensing and gating behavior of ERG channels in vivo, we altered critical amino acids in the conserved transmembrane domain by introducing

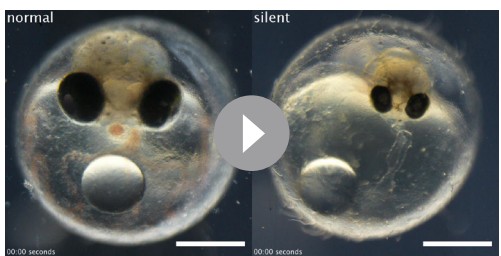

**Video 1.** evoBE4max introduced premature STOP codon in *O.latipes tnnt2a* results in silent heart phenotype. Time-lapse movie (10 seconds) of the beating medaka heart. Scale bar = 400 µm.
https://elifesciences.org/articles/72124/figures#video1

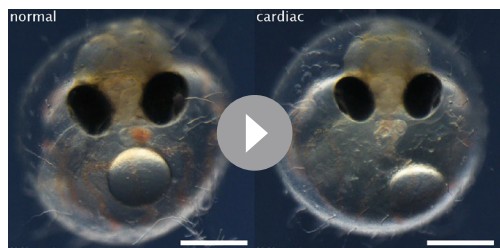

**Video 2.** evoBE4max introduced premature STOP codon in *O.latipes kcnh6a* results in ventricular asystole accompanied by morphological alterations. Time-lapse movie (10 seconds) of the beating medaka heart. Scale bar = 400 µm.
https://elifesciences.org/articles/72124/figures#video2

precise point mutations (*Figure 6—figure supplement 1a-b*), modeled on alleles described in zebrafish (*Arnaout et al., 2007*).

We used ACEofBASEs to select sgRNAs that facilitate base edits targeting and putatively inactivating the S4 and S4-S5 linker domain of the medaka ERG channel, resulting in the sgRNAs *kcnh6a-R512*, *kcnh6a-T507*, and *kcnh6a*-D521 (*Figure 6—figure supplement 1a-b*). We addressed the consequences of editing crucial amino acids in combination and individually.

Co-injection of all three sgRNAs with ABE8e editor mRNA into medaka embryos revealed a high frequency of specific heart phenotypes at 4 dpf (82% and 88% respectively, two independent biological replicates; *Figure 6c*, *Figure 6—figure supplement 4*), including significantly reduced or complete loss of ventricular contraction (*Figure 6b–c*, *Video 3*).

The introduction of a missense mutation to exchange a single arginine at *kcnh6a-R512* was particularly interesting, since the edit of a single nucleotide removes a positive charge at a crucial position. Injection of the sgRNA *kcnh6a-R512* together with ABE8e editor mRNA into medaka embryos demonstrated that this S4 sensor charge is functionally essential in vivo. Its exchange resulted in partially or entirely silent ventricles in the majority of scored F0 embryos (*Figure 6c*, *Figure 6—figure supplement 4*) as suggested by electrophysiological studies on heterologously expressed *Hs* ERG (*Zhang et al., 2004*).

Analysis of the genomic *kcnh6a* locus around R512 of pooled (*kcnh6a-T507, -R512, -D521*) and *kcnh6a-R512* single editants revealed reproducibly high conversion rates of A8-to-G resulting in p.R512G (92.1% ± 6.6% and 91.7% ± 9.5%), respectively (*Figure 6d*). In pooled editants the two other sgRNA sites T507 and D521 also reached high editing efficiencies of 81.6% ± 7.2% (A6-to-G) and 61.0% ± 12.5% (A5-to-G), respectively (*Figure 6d'–d''*; *Supplementary file 2*). Notably, ABE8e-mediated adenine base editing achieved homozygosity in three out of four imaged R512G embryos in F0 (*Figure 6e*) with Amplicon-seq revealing indel rates ranging from 0.8% to 27.8% in different individuals (*Figure 6f*, *Figure 6—figure supplement 5*).

Investigating the ABE8e-mediated R512G missense editants in a *myl7::GFP* reporter (*Gierten et al., 2020*) revealed a striking morphological impact of the primary repolarization phenotype in specimens with silent hearts. While atrioventricular differentiation is complete, the ventricular muscle showed major growth and differentiation deficiencies. High-resolution imaging of silent ventricles showed ventricular collapse and multiple vesicle-shaped, aneurysm-like structures (*Figure 6g*). Whether this structural phenotype directly relates to *kcnh6a* effects or, more likely, is a secondary consequence of the lack of forces generated by chamber contractions and directed blood flow is an exciting starting point for future investigations.

Taken together, applying ABE8e in medaka embryos enabled modifying a conserved voltage sensor domain of the LQTS/SQTS-related potassium channel ERG in vivo at the level of a single amino acid in F0.

## Functional validation of congenital heart disease-associated missense mutations with SNV resolution using cytosine editing

We have demonstrated the functionality and potential of CBEs and ABEs by introducing PTCs or missense mutations into well-studied genes with reference phenotypes. We next used base editing to validate specific mutations in novel candidate genes associated with congenital heart disease. Extensive sequencing studies have revealed a polygenic origin of CHD, uncovering a plethora of candidate genes with enrichment of SNVs with predicted pathogenic relevance acting as protein-truncating or missense variants (*Homsy et al., 2015*; *Sifrim et al., 2016*; *Zaidi et al., 2013*). To validate novel SNVs using base editing, we chose four highly ranked candidate genes, *DAPK3*, *UBE2B*, *USP44*, and *PTPN11*, each with a specific CHD-associated missense mutation (*Homsy et al., 2015*; *Zaidi et al., 2013*).

Using the missense mutation loci, ACEofBASEs identified sgRNAs compatible with cytosine editing to introduce missense mutations in the highly conserved medaka orthologues resulting in identical or alternative amino acid changes (*Figure 7a*, *Figure 7—figure supplement 1*).

Each of these sgRNAs was co-injected with evoBE4max into medaka zygotes using wild-type strains Cab or HdrR with dual-color cytosolic and nuclear heart-specific labels (*myl7::EGFP, myl7::H2A-mCherry*). Individual embryos were initially subjected to Sanger sequencing and EditR analysis, quantifying C-to-T transition. For each of the four genes we obtained the expected edits with 86.9% ±

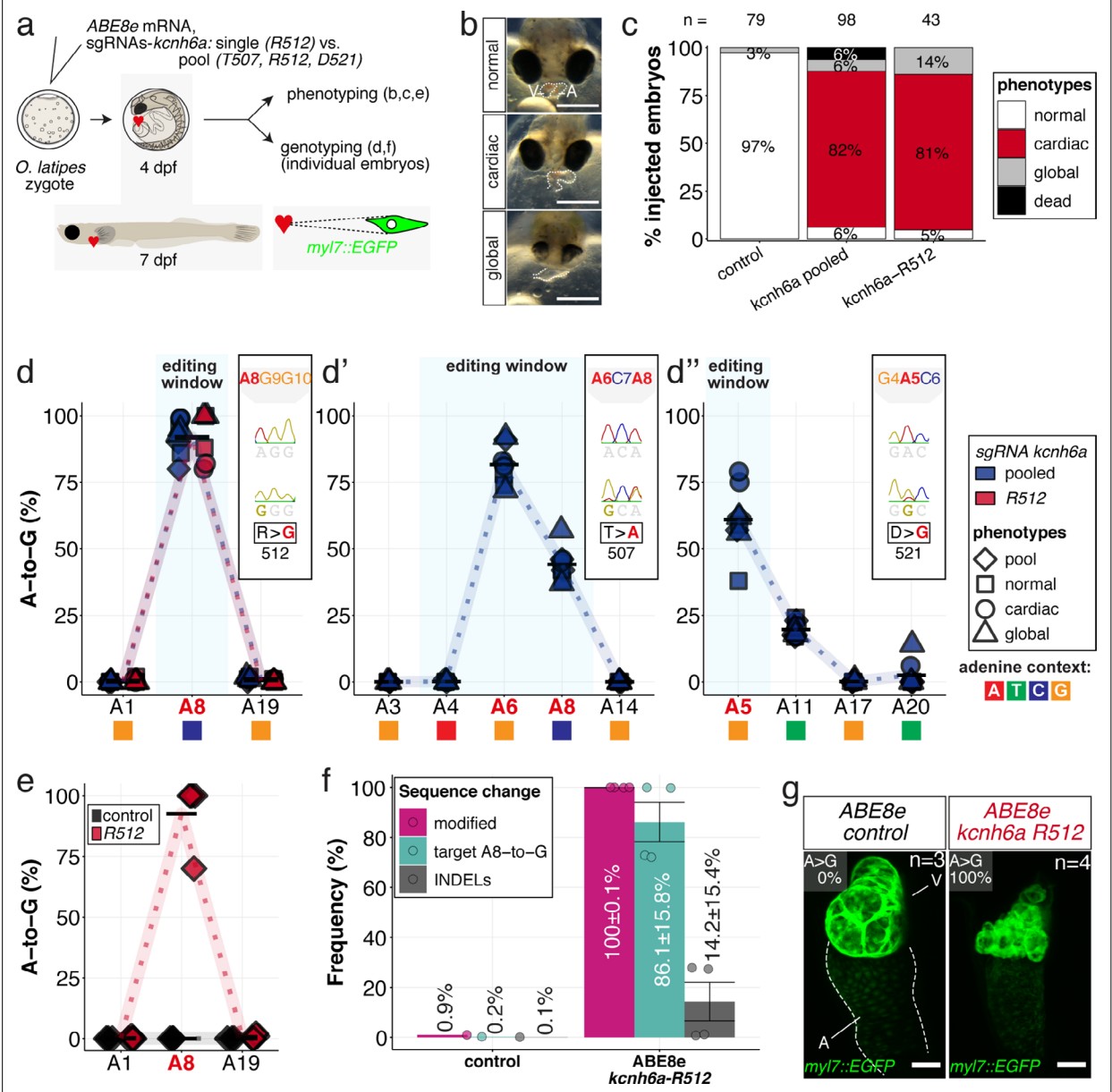

**Figure 6.** In vivo modeling of human LQTS-associated mutations using adenine base editing of the medaka ERG channel gene *kcnh6a*. (**a**) Regime of ABE8e mRNA injections with a single (*kcnh6a-R512*) or pooled sgRNAs (*kcnh6a-T507, -R512, -D521*) targeting different amino acid codons in the voltage sensor S4 domain/S4-S5 linker of the medaka potassium channel ERG in *myl7::EGFP* (*Cab* strain) transgenic embryos; control injection included ABE8e mRNA only. (**b**) Phenotypes in F0 comprised primary cardiac malformation (dysmorphic ventricle with impaired contractility) and more severe global phenotypes with general retarded development and prominently dysmorphic hearts the proportions of which are given in (**c**). Scale bar = 400 μm. (**d-d''**) Genotyping summaries of the three sgRNA loci with phenotype class annotations for each genotyped specimen with a comparison of single sgRNA-R512 injection to a pool with two additional sgRNAs (T507 and D521) targeting the medaka ERG S4 voltage sensor; inlets display Sanger reads with the editing of A8 (**d**), A6 and A8 (**d'**) and A5 (**d''**) contained in the core editing windows; sgRNA pool (n = 8) and sgRNA-R512 (n = 6). To highlight the dinucleotide context, the nucleotide preceding the target A is shown by red (**A**), green (**T**), blue (**C**) and yellow (**G**) squares below the respective A. (**e–g**) Confocal microscopy of the heart in a *myl7::EGFP* reporter line injected with ABE8e mRNA and sgRNA-R512 at 7 dpf reveals significant chamber wall defects of non-contractile/spastic ventricles with A-to-G editing of 100% in 3/4 of the specimen as determined by Sanger sequencing (**e**). (**f**) Amplicon-seq of the same gDNA samples (single embryos, n = 4) quantified target A8-to-G editing and indel frequencies. Aligned Illumina-reads analyzed, 11,387 (control); 24,936 (ABE8e rep1); 4038 (ABE8e rep2); 75,148 (ABE8e rep3); 86,327 (ABE8e rep4). Images show maximum z-projections of optical slices acquired with a z-step size of 1 μm (**g**). Note the display of A-to-G conversion rates. Scale bar = 50 μm. V = ventricle, A = atrium, dpf = days post fertilization.

The online version of this article includes the following figure supplement(s) for figure 6:

*Figure 6 continued on next page*

Figure 6 continued

**Figure supplement 1.** Robustness of interrogating gene function by introducing stop-gain mutations in medaka demonstrated at the N-terminal *kcnh6a-Q11* locus in F0 through CBEs.

**Figure supplement 2.** evoBE4max cytosine base editing enables efficient installation of PTCs or missense mutations at three additional *kcnh6a* loci.

**Figure supplement 3.** Analysis of BE4-Gam mediated PTC installation in *kcnh6a* in F1.

**Figure supplement 4.** Targeting the three *kcnh6a* loci simultaneously or *kcnh6a-R512* alone in a different genetic background (*myl7::EGFP, myl7::H2A-mCherry; HdrR* strain) recapitulates phenotypic proportions.

**Figure supplement 5.** Sequence composition determined by Amplicon-seq of ABE8e *kcnh6a-R512* editants surrounding the *kcnh6a-R512* sgRNA target site ± 5 bp.

8.7% (dapk3-p.P204L), 54.2% ± 24.8% (ube2b-p.R8Q), 99.7% ± 0.9% (usp44-p.E68K), and 88.0 ± 7.3 (ptpn11-p.G504E/K) average efficiency, respectively (*Figure 7*, *Figure 7—figure supplement 2*, *Supplementary file 3*). In addition, missense bystander edits were observed in *dapk3* (p.L205F, 61.7% ± 14.1%), *ube2b* (p.R7K, 67.4% ± 16.8%), and *usp44* (p.M67I, 97.0% ± 5.0%). Corresponding Amplicon-seq data for *ube2b* experiments revealed slightly lower rates of the intended edits of 46.6% ± 11.9% and low indel rates (4.9% ± 4.3%) (*Figure 7—figure supplement 3*).

The first time point of phenotypic analysis at 4 dpf revealed cardiovascular phenotypes for all four CBE-driven missense mutations (*Figure 7d*). While atrium and ventricle were specified, we observed a lack of normal dextra heart looping (R-loop) with a displacement of the ventricle to the left (L-loop, left-right asymmetry defect), vessel malformation and blood clotting (*dapk3*), defective looping, and aberrant heart morphology (*ube2b*, *usp44*), and mesocardia, that is, lack of looping, and smaller ventricles (*ptpn11*) (*Figure 7*, *Figure 7—figure supplement 4*). Notably, missense mutations in each of the four candidate genes specifically impacted the cardiovascular system (*Figure 7—figure supplement 4*), validating their suspected impact.

Representative embryos were further characterized by confocal microscopy of the heart at 7 dpf. High-resolution imaging revealed abnormal AV channel formation, dysmorphic atrium and ventricle in the *usp44*-p.E68K editants (100% edited allele, *Figure 7f*). The *ptpn11*-p.G503E editants (88% edited allele) showed a small ventricle and linear atrium (*Figure 7g*). Additionally, *ube2b* and *dapk3* editants displayed consistent cardiac phenotypes (*Figure 7—figure supplement 5*).

Finally, to address the contribution of the edited nucleotides resulting in missense codons in *dapk3* and *usp44* to heart development, we correlated the F1 phenotypes and the respective genotypes (*Figure 7h–i*).

To establish heterozygous individuals, editing was performed at reduced concentrations of the respective base editors employed, analogous to the approach described above for *tnnt2a* (*Figure 5f*). Those were crossed and the resulting phenotypes in the F1 offspring were characterized and correlated individually to the underlying genotype as revealed by Sanger sequencing.

For *dapk3* the F1 mutant analysis revealed that the single targeted amino acid change P204L resulted in cardiac abnormalities (looping defects) comparable to those already observed in the F0 editants. This F1 analysis also uncovered that the L205F bystander edits had similar phenotypic consequences and caused looping defects with varying severity (*Figure 7h*, *Figure 7—figure supplement 6*).

Interestingly, the range of phenotypes observed for individuals carrying the usp44-E68K allele initially scored in the F0 editants was also apparent in the F1 homozygous mutants, highlighting an incomplete penetrance of this missense mutation. Since other missense mutations at this site (e.g. M67I) also displayed an incomplete penetrance with respect to cardiac phenotypes (ranging from looping defect via aberrant heart morphology to global developmental defects, particularly of the brain and the eyes), the altered function of the *usp44* mutant appears variably

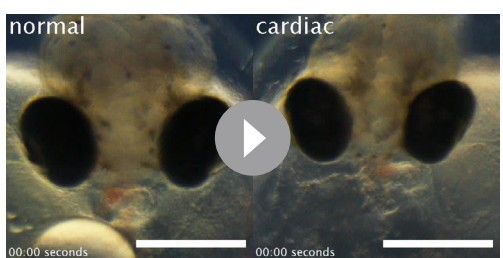

**Video 3.** ABE8e driven installation of the R512G missense mutation in *O.latipes kcnh6a* results in ventricular asystole accompanied by morphological alterations. Time-lapse movie (10 s) of the beating medaka heart. Scale bar = 400 μm.

https://elifesciences.org/articles/72124/figures#video3

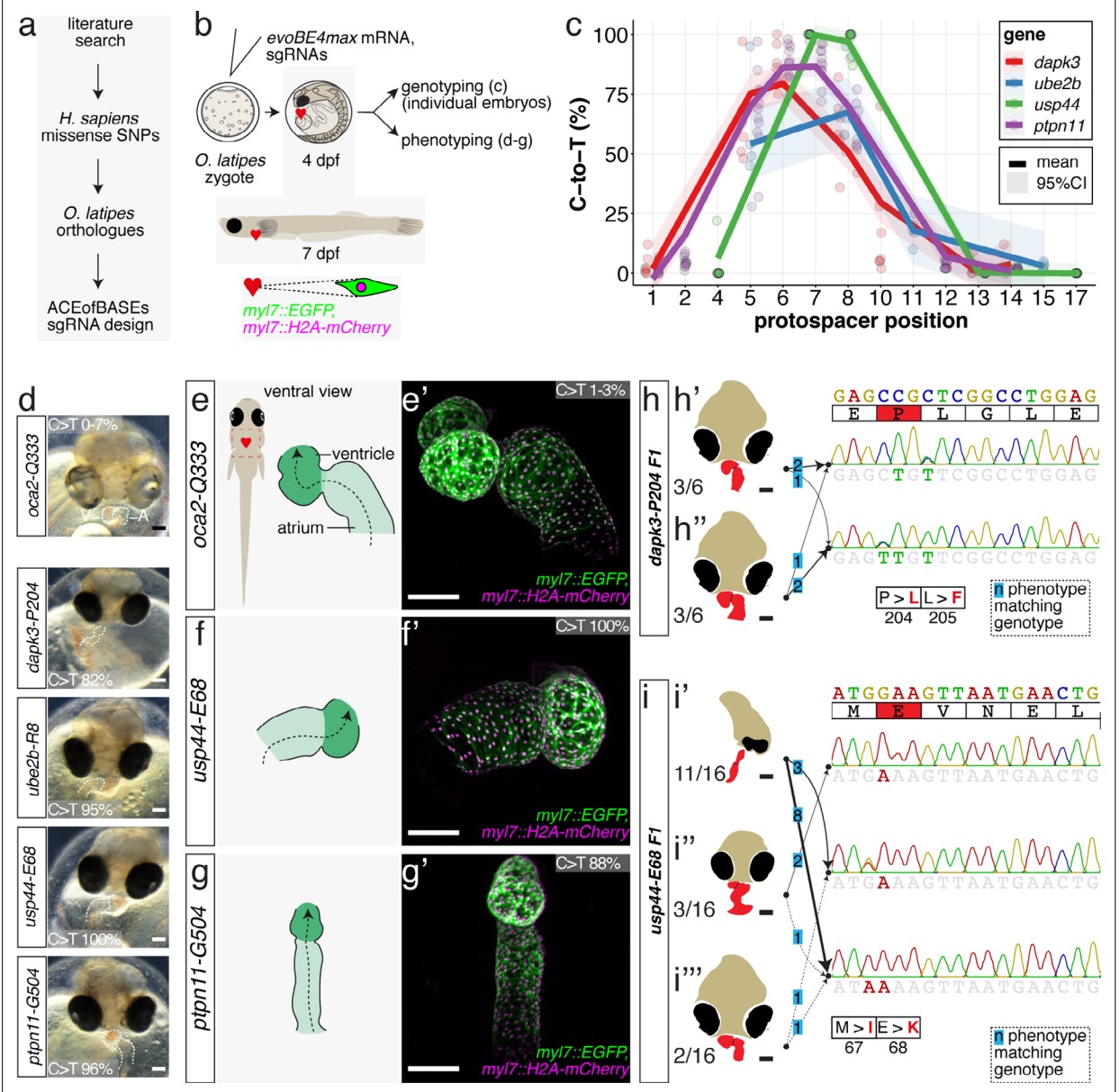

**Figure 7.** Cytosine base editing enables human CVD-associated SNV validation. (**a**) Candidate human CVD gene SNV validation workflow. (**b**) To target the SNVs *evoBE4max mRNA* was co-injected into the 1 cell stage of the medaka wild-type or *myl7::EGFP, myl7::H2A-mCherry* reporter strain together with the corresponding target or *oca2-Q333* (control) sgRNAs. Individual, imaged, embryos were then further analyzed to determine the rate of C-to-T transversions. (**c**) Cytosine editing efficiencies are substantial for all candidate genes tested. Data shown in *Figure 7—figure supplement 2* was replotted, including all data points from a-d across all target cytosine along the protospacer. Sample numbers: *dapk3-P204* (n = 7), *ube2b-R8* (n = 5), *usp44-E68* (n = 11), and *ptpn11-G504* (n = 11). (**d**) Representative phenotypes of 4 dpf base edited embryos are shown for all four tested candidate CVD genes including *oca2-Q333* controls. Top, ventral view, with V = ventricle, A = atrium. (**e–g**) Confocal microscopy of selected candidate validations in the reporter background. Hearts were imaged in 7 dpf hatched double fluorescent embryos. Images show maximum projections of the entire detectable cardiac volume with a step size of 1 μm. Cartoons (left) highlight the looping defects observed in *usp44* and *ptpn11* base edited embryos with ventricle-atrium inversion (**f**) or tubular heart (**g**). (**e′-g′**) Imaged embryos were subsequently genotyped and quantified C-to-T transversions for the target codon are shown. Note: due to the inverted nature of the confocal microscope used, raw images display a mirroring of observed structures, which we corrected here for simpler appreciation. Phenotypic analysis of F1 *dapk3-P204L* (**h**) and *usp44-E68K* (**i**) embryos revealed that homozygous changes at P204L or E68K lead to cardiac malformations with varying degree: looping (**h′**) and mild looping defects (**h″, i‴**); altered heart morphology (**i″**). Bystander edits (hetero- or homozygous, usp44-E68K) lead to additional developmental defects, including brain and eye abnormalities (**i′**). Scale bar = 100 μm (**d, e–i**). dpf = days post fertilization.

The online version of this article includes the following figure supplement(s) for figure 7:

*Figure 7 continued on next page*

*Figure 7 continued*

**Figure supplement 1.** CVD-associated SNVs can be mapped to the orthologous medaka peptide sequence with high conservation and are expressed during heart development.

**Figure supplement 2.** Cytosine base editing allows the introduction of human CVD-associated missense mutations in medaka in F0.

**Figure supplement 3.** Amplicon-seq of *ube2b-R8* evoBE4max editants.

**Figure supplement 4.** Phenotypic categorization of cytosine base edited embryos in medaka in F0.

**Figure supplement 5.** Confocal microscopy of evoBE4max validated CVD genes.

**Figure supplement 6.** Phenotype-genotype correlation of dapk3-P204L/L205F embryos with mild (**c**) and moderate looping defects (**d**).

**Figure supplement 7.** Phenotype-genotype correlation of usp44-M67I/E68K embryos with mild looping defects (**c**), altered heart morphology with slight developmental delay (**d**) and severe alterations of heart morphology with concomitant strong global developmental defects (**e**).

compensated. Although specific genotypes resulted in the variable expression of phenotypes, the accumulation of both missense mutations seems to aggravate the phenotypic impact resulting in a higher fraction of globally affected embryos (*Figure 7i*, *Figure 7—figure supplement 7*).

The F1 results demonstrate that F0 base editing screening efficiently indicated the cardiovascular significance of disease-associated SNVs.

In summary, we have established and validated a comprehensive toolbox for the context-dependent editing of single nucleotides in model systems. We transferred human mutations to medaka to translate an association into causality. We applied base editing in medaka and precisely modeled single SNVs in vivo, validating single missense mutations initially associated with human congenital heart defects and highlighting the potential of targeted base editing for disease modeling.

## Discussion

Determining the in vivo functional consequences of SNVs associated with physiological trait variation and disease in humans is critical to assess causative genetic variants. We provide a framework for in vivo base editing to close the gap between SNV discovery and validation using small fish model systems, medaka and zebrafish, allowing accurate phenotype assessment of gene function interference. Our online base editing tool ACEofBASEs allowed us to probe the efficiency of current CBEs and one ABE at various test loci in fish and to uncover the developmental impact of point mutations in four novel CHD candidate genes *dapk3*, *ube2b*, *ups44, and ptpn11*. Already in F0, editants display robust and conclusive phenotypes that were confirmed in homozygous F1 embryos. Our analyses indicate that the framework presented provides all means to rapidly address a larger number of individual base changes efficiently identifying relevant sites already in the injected generation.

The synergy between CBEs and ABEs and our open-access web tool ACEofBASEs facilitates immediate access to base editing experiments in cell- and organism-based assays. The modular design of our software allows entry- as well as expert-level base editing applications. In contrast to existing tools like the BE-designer (*Hwang et al., 2018*) or PnB Designer (*Siegner et al., 2021*) or custom scripts to generate nonsense mutations (*Rosello et al., 2021b*), ACEofBASEs presents the altered amino acid composition, direct off-target assessment with linked sequence information, and details for sgRNA cloning. In addition, comprehensive editing annotations, a selection of base editor PAM variants, and genome selection available for an extensive (and constantly growing) collection of species expand the applicability of base editing to laboratories working on a wide range of different model systems.

ACEofBASEs guided immediate and efficient base editing at multiple relevant loci. We used it to assess the in vivo performance of three recent state-of-the-art CBEs, BE4-Gam, ancBE4max, evoBE4max, and ABE8e, in comparison to the respective in vitro specifications. Our results show comparable overall editing efficiencies for ancBE4max, evoBE4max, and ABE8e, with the highest level of edits at efficiencies ranging from 94% to 100%, indicating almost quantitative bi-allelic editing in the injected generation (*Supplementary files 1-3*). We confirmed the high performance of the applied base editors by additionally examining both cytosine and adenine base editing outcomes with deep Amplicon-sequencing (Illumina sequencing), each at three loci (*Supplementary file 4*). The averaged editing efficiency determined by Sanger vs. Illumina sequencing of the identical samples was 77.8 ± 20.8% vs 71.7 ± 22.3% for evoBE4max and 92.9 ± 3.7% vs 83.7 ± 3.9% for ABE8e. Considering the indel frequencies determined by Illumina-sequencing, 9.0% ± 4.9% and 12.7% ± 4.1%, respectively

for evoBE4max and ABE8e, the sequence validation method of choice will depend on the biological question and the state of the experiment. Amplicon-seq provides insight into the allele distribution and allows estimating the frequency of the intended edit, which is important to ensure the presence of the intended edit if it is rare. Rapid Sanger sequencing on the other hand overestimates the editing frequency at the intended position and is limited in detecting low-to-moderate levels of indels. If the overall percentage of modified alleles is close to 100%, moderate level of indels do not impact on the overall conclusion of the phenotypic outcomes, given the prevalence of the intended editing event.

We have determined the in vivo editing windows and context-dependent efficiencies of all four editors (*Figure 8a–a'*; *Table 1*). Our in vivo findings confirm the in vitro characteristics of the APOBEC-1 deaminase-inherent canonical dinucleotide context sequence preference (TC >CC > AC > GC), which was ameliorated in ancBE4max and evoBE4max, both with improved editing at AC or GC-dinucleotides, respectively (*Arbab et al., 2020*; *Huang et al., 2021*; *Komor et al., 2016*). Similarly and consistent with the in vitro observations (*Richter et al., 2020*) a dinucleotide preference was not observed for ABE8e (*Figure 8b–b'*).

Our analyses show enhanced efficiencies compared to previous reports for cytosine base editing in fish (*Rosello et al., 2021b*; *Zhang et al., 2017*; *Zhao et al., 2020*). In particular, the adenine base editor variant ABE8e in both medaka and zebrafish remarkably exceeded efficiencies previously reported for ABE7.10 in zebrafish (*Qin et al., 2018*). Overall, high efficiencies and precision in vivo with limited bystander edits and undetectable off-target DNA editing allowed to overcome the bottleneck of current genome engineering strategies for the reliable, somatic interrogation of DNA variants.

We demonstrate the power of in vivo F0 analysis and validation of human genetic variants in fish by generating PTCs and specific missense mutations in essential cardiac genes resulting in phenocopies of the respective reference heart phenotypes and matched F1 phenotypes. Exchanging a single positively charged acid to a neutral one in the ultra-conserved voltage sensor domain (S4) of the potassium channel ERG (*kcnh6a*) resulted in the formation of a non-contractile ventricle equal to previously reported phenotypes of null mutants established in zebrafish (*Hoshijima et al., 2016*). Given the high editing efficiency, the ventricle collapse with secondary morphological aberrations revealed by high-resolution imaging highlights a previously unreported ERG functionality consistently observed in the injected generation. In contrast to the knockin-knockout strategy (*Hoshijima et al., 2016*), the base editing approach allows to efficiently interrogate distinct human point mutations in potentially disease-relevant genes as well as addressing structure function relationships by editing individual amino acids. This is particularly appealing to address clinically relevant mutations in ERG to investigate acquired LQTS by drug exposure or, vice versa, the pharmacological control of heritable LQTS.

Precise base editing in medaka pinpointed the developmental impact of novel missense mutations in four genes, *DAPK3*, *UBE2B*, *USP44*, and *PTPN11*. Those were associated with structural CHD in parent-offspring trio exome sequencing studies (*Homsy et al., 2015*; *Zaidi et al., 2013*). We accurately modeled these human de novo mutations guided by ACEofBASEs and applied cytosine base editing to the conserved medaka orthologs.

While missense mutations in the kinase domain of *DAPK3* had been observed in various carcinomas (*Brognard et al., 2011*), there was only circumstantial evidence linking it to the heart. In biochemical analyses with isolated proteins, DAPK3 could phosphorylate the regulatory light chain associated with cardiac myosin (MYL2) (*Chang et al., 2010*). Our base editing experiment provides the first functional evidence linking the missense mutation p.P193L, associated with conotruncal defects in humans (*Zaidi et al., 2013*), to CHD. Introduction of the p.P204L mutation into the corresponding position of the dapk3 medaka ortholog resulted in looping and morphological heart defects during embryonic development, which we confirmed in homozygous F1 embryos.

Our base editing experiments modeling patient-based mutations to UBE2B and USP44, both involved in the post-translational modifications of histones (H2Bub1), show their essential role in heart looping. While the knockdown of ube2b in *Xenopus* did not result in an apparent phenotype (*Robson et al., 2019*), mutations in the gene were correlated in CHD patients. Strikingly, the introduction of the single p.R8Q missense mutation into the medaka *ube2b* ortholog resulted in aberrant L-loop phenotypes in medaka embryos.

USP44 regulates the cell cycle as deubiquitinase acting on H2Bub1 during embryonic stem cell differentiation (*Fuchs et al., 2012*; *Stegmeier et al., 2007*). Respective mouse knockout models

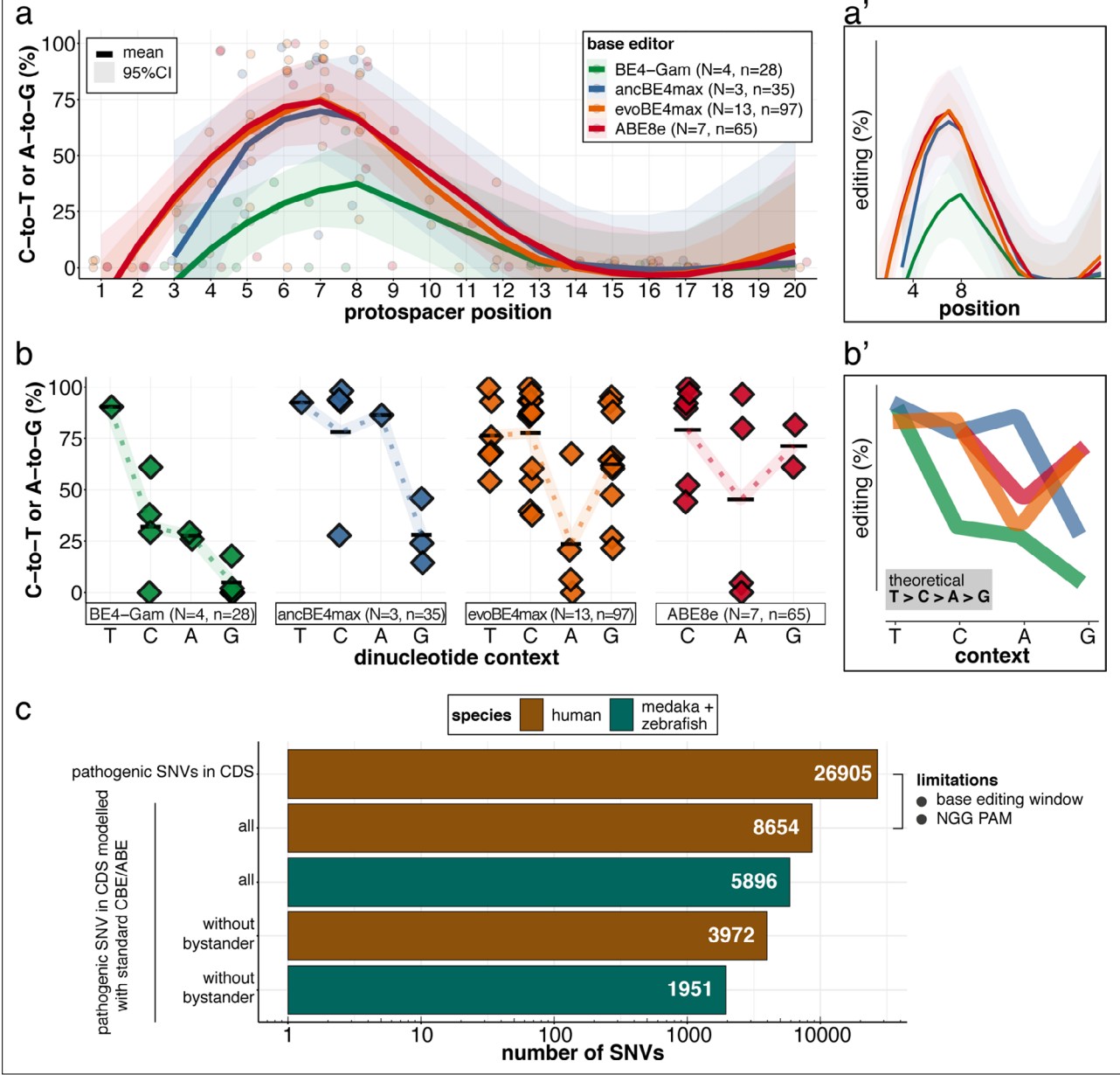

**Figure 8.** Recapitulation of in vitro base editing characteristics combined with a plethora of conserved variants make fish excellent models to validate human pathogenic SNVs. (**a**) Transition efficiencies for CBEs and ABE8e tested in this study in medaka across the entire protospacer are shown as mean with 95% confidence interval (CI). Each data point represents the mean efficiency for the locus (sgRNA) tested. N = represents the number of loci tested with the respective editor. n = total number of genomes used for quantification of editing efficiencies (a pool of five embryos was counted as five genomes). (**a′**) Simplified scheme only showing mean and CI. (**b**) Summary of dinucleotide preference for the tested base editors, calculated for the standard editing window (4-8). Each data point represents the mean editing efficiency of the corresponding editor for a particular protospacer position. (**b′**) Simplified scheme of overlaid dinucleotide logic. (**c**) Analysis of human pathogenic CDS SNVs annotated in ClinVar reveals that a remarkable portion of these SNVs have orthologous sequences in medaka or zebrafish that can be mimicked by CBEs or ABEs following editing window (4-8) and NGG PAM restrictions. Modeling SNVs mutations may be achieved with stringent criteria (no bystander mutations accepted, n = 1951) or less stringent selection (allowing bystander mutations 'all', n = 5896). Note: the number of SNVs shown for medaka + zebrafish, corresponds to a set-up in which these species are complementing each other.

hint at a tumor suppressor role (*Zhang et al., 2012*). Interestingly, a specific glutamate residue (Ol E68, Hs E71) was associated with CHD in GWAS (*Zaidi et al., 2013*), demanding a precise editing for functional validation that we applied in the medaka ortholog at the corresponding position. This allowed to disentangle the phenotypic range observed in F0 and similarly in F1 *usp44* null mutants

**Table 1.** Estimated editing windows and dinucleotide preference affecting editing efficiencies. Comparison of literature estimates (in vitro) and in vivo metrics observed in this study.

| Base editor | Highest average editing efficiency (site tested) | Editing window on protospacer: overall (peak) activity | | Dinucleotide sequence preference | |
|---|---|---|---|---|---|
| | | In vitro[*] | This study | In vitro[*] | This study |
| BE4-Gam | 61.0 ± 10.4 (*kcnh6a-p.Q11X*) | 3–10 (4-8) | 4–8 (NA) | canonical | canonical |
| ancBE4max | 93.8% ± 7.9% (*oca2-p.Q333X*) | 3–9 (4-7) | NA (5-8) | << G<u>C</u> | << G<u>C</u> |
| evoBE4max | 99.7% ± 0.9% (*usp44-p.E68K*) | 1–11 (4-8) | 3–12 (5-8) | << A<u>C</u> | << A<u>C</u> |
| ABE8e | 100% (*oca2-p.Q256R*) | 3–11 (4-8) | 3–11 (4-8) | - | < A<u>C</u> |

NA – not sufficient data to estimate.

[*]*Arbab et al., 2020*; *Huang et al., 2021*; *Richter et al., 2020*.

and emphasizes the physiological relevance of a single amino acid for the proper development of the heart.

Previous experiments in zebrafish (mRNA overexpression of pathogenic *PTPN11* variants, *Bonetti et al., 2014*) had argued for a potential role of *PTPN11* in cardiac development. Engineering the PTPN11 missense mutation p.G503E residing in the protein tyrosine phosphatase (PTP) domain (*Homsy et al., 2015*) into the corresponding position of the medaka ortholog resulted in impaired cardiac looping, demonstrating that this amino acid is essential for proper PTPN11 function in cardiac development.

Those four examples highlight the power of the combination of ACEofBASEs as a prediction tool and the respective base editors to instantly validate associations from human datasets. This allowed the immediate establishment of highly informative small animal models for human diseases that could be queried already in the F0 generation, underpinning the power of the approach in addressing the pathophysiology of human disease-associated DNA variants.

Although the list of successfully in vivo validated SNVs remains limited (*Claussnitzer et al., 2020*), the combined CBE-ABE (standard editing window, NGG PAM) action provides access to almost 30% of human variants in ClinVar (*Gaudelli et al., 2017*; *Komor et al., 2016*; *Landrum et al., 2016*) for functional interrogation. We analyzed the pathogenic ClinVar SNVs' conservation revealing that CBEs and ABEs can directly model 5896 human CDS SNVs in the orthologous medaka and zebrafish genes combined, consistent with the high fraction of conserved orthologs in zebrafish (*Howe et al., 2013*) and medaka (*Kasahara et al., 2007*). Notably, 1951 pathogenic CDS SNVs can be modeled without bystander mutations (*Figure 7c*). The practical use of CBE and ABE Cas variants with broadened PAM compatibilities, such as xCas9 or SpRY, is estimated to collectively expand variant validation possibilities, covering up to 95% of pathogenic ClinVar transition mutations (*Anzalone et al., 2020*; *Hu et al., 2018*; *Walton et al., 2020*). A recent report of NR PAM cytosine base editing in fish (*Rosello et al., 2021a*) opens the door for future validation experiments using expanded PAM editors. With the recent advances in the evolution of deaminases (*Chen et al., 2021*; *Koblan et al., 2021*; *Kurt et al., 2021*; *Zhao et al., 2021*), the substitution flexibility is constantly extended.

Given the high precision and efficiency, the parallel introduction of multiple edits is within reach in the near future. This will ultimately allow tackling polygenic traits in the organismal context. It will likewise provide experimental access to the regulatory genome, as recently demonstrated by combining epigenomics with adenine base editing in vitro that mapped and validated sickle cell disease-associated cis-regulatory elements of the fetal haemoglobin (*Cheng et al., 2021*).

In conclusion, we demonstrate that in vivo base editing employing our online tool ACEofBASEs, is instantly applicable to a wide range of developmental and genetic disease studies in native genomic contexts. Validation of the plethora of genetic variants surfaced by large-scale sequencing in humans, which require effective functional testing strategies, can now be immediately addressed by ACEofBASEs guided in vivo base editing in fish. Future studies in model organisms focusing on the pre-clinical examination of genetic disease variants will help direct the translational discovery process to improve personalized clinical care of patients with individual genetic variant profiles.

# Materials and methods

**Key resources table**

| Reagent type (species) or resource | Designation | Source or reference | Identifiers | Additional information |
|---|---|---|---|---|
| Strain, strain background (Oryzias latipes) | Cab | *Loosli et al., 2000* | N/A | medaka Southern wild type population; Wittbrodt lab |
| Strain, strain background (Oryzias latipes) | Cab (myl7::EGFP) | *Gierten et al., 2020* | N/A | Wittbrodt lab |
| Strain, strain background (Oryzias latipes) | HdrR (myl7::EGFP myl7::H2A-mCherry) | *Hammouda et al., 2021* | N/A | Wittbrodt lab |
| Strain, strain background (Danio rerio) | AB | ZIRC | ZFIN: ZBD-GENO-960809–7, RRID:ZIRC_ZL1 | Wildtype zebrafish strain |
| Recombinant DNA reagent | pGGEV_4_BE4-Gam (plasmid) | *Thumberger et al., 2022* | N/A | Plasmid vector for in vitro transcription of BE4-Gam mRNA; Wittbrodt lab |
| Recombinant DNA reagent | pCMV_AncBE4max, (plasmid) | Addgene | Addgene plasmid #112094; http://n2t.net/addgene:112094; RRID:Addgene_112094 | Plasmid vector for expression or in vitro transcription of ancBE4max mRNA from pCMV; Liu lab |
| Recombinant DNA reagent | pBT281(evoAPOBEC1-BE4max) (plasmid) | Addgene | Addgene plasmid #122611; http://n2t.net/addgene:122611; RRID:Addgene_122611 | Plasmid vector for expression of evoBE4max in mammalian cells; Liu lab |
| Recombinant DNA reagent | pCS2+ (plasmid) | *Rupp et al., 1994* | Xenbase: XB-VEC-1221270 | high-level transient expression for mRNA synthesis and injection numerous aquatic organisms |
| Recombinant DNA reagent | pCS2+_evoBE4max | This paper | N/A | See Materials and methods |
| Recombinant DNA reagent | pBABE8e (plasmid) | Addgene | Addgene plasmid #138489; http://n2t.net/addgene:138489; RRID:Addgene_138489 | Plasmid vector for expression or in vitro transcription of ABE8e mRNA from pCMV; Liu lab |
| Recombinant DNA reagent | DR274 (plasmid) | Addgene | Addgene plasmid #42250; http://n2t.net/addgene:42250; RRID:Addgene_42250 | sgRNA expression vector to create sgRNA to a specific sequence with T7 promoter for in vitro transcription |
| Recombinant DNA reagent | oligonucleotides | Eurofins Genomics | PCR primers, sgRNA cloning primers, sequencing primers | see Materials and Methods |
| Gene (Oryzias latipes) | oca2 | Ensemble genome browser | Ensemble (release 96): ENSORLG00000015893 CDS | Medaka OCA2 melanosomal transmembrane protein |
| Gene (Danio rerio) | oca2 | Ensemble genome browser | Ensemble (release 103): ENSDARG00000061303.8 CDS | Zebrafish OCA2 melanosomal transmembrane protein |
| Gene (Homo sapiens) | oca2 | Ensemble genome browser | Ensemble (release 103): ENSG00000104044.16 CDS | OCA2 melanosomal transmembrane protein |
| Gene (Oryzias latipes) | tnnt2a | Ensemble genome browser | Ensemble (release 95): ENSORLG00000024544.1 CDS | Medaka cardiac muscle-like troponin T |
| Gene (Oryzias latipes) | kcnh6a | Ensemble genome browser | Ensemble (release 93): ENSORLG00000002317.1 CDS | Encodes *Ol* ERG |
| Gene (Danio rerio) | kcnh6a | Ensemble genome browser | Ensemble (release 103): ENSDARG00000001803.12 CDS | Encodes *Dr* ERG |
| Gene (Homo sapiens) | kcnh2 | Ensemble genome browser | Ensemble (release 103): ENSG00000055118.16 CDS | Encodes *Hs* ERG |
| Gene (Oryzias latipes) | tnnt2c | Ensemble genome browser | Ensemble (release 93): ENSORLG00000016386.1 CDS | Medaka cardiac troponin T2c |

*Continued on next page*

*Continued*

| Reagent type (species) or resource | Designation | Source or reference | Identifiers | Additional information |
|---|---|---|---|---|
| Gene (*Oryzias latipes*) | *s1pr2* | Ensemble genome browser | Ensemble (release 93): ENSORLG00000005560.1 CDS | Medaka sphingosine-1-phosphate receptor 2 |
| Gene (*Oryzias latipes*) | *dapk3* | Ensemble genome browser | Ensemble (release 103): ENSORLG00000017965.2 CDS | Medaka death associated protein kinase 3 |
| Gene (*Homo sapiens*) | *dapk3* | Ensemble genome browser | Ensemble (release 103): ENSG00000167657.14 CDS | Human death associated protein kinase 3 |
| Gene (*Oryzias latipes*) | *ube2b* | Ensemble genome browser | Ensemble (release 103): ENSORLG00000000951.2 CDS | Medaka ubiquitin conjugating enzyme E2 B |
| Gene (*Homo sapiens*) | *ube2b* | Ensemble genome browser | Ensemble (release 103): ENSG00000119048, CCDS4174 | Human ubiquitin conjugating enzyme E2 B |
| Gene (*Oryzias latipes*) | *usp44* | Ensemble genome browser | Ensemble (release 103): ENSORLG00000016627.3 CDS | Medaka ubiquitin specific peptidase 44 |
| Gene (*Homo sapiens*) | *usp44* | Ensemble genome browser | Ensemble (release 103): ENSG00000136014.12 CDS | Human ubiquitin specific peptidase 44 |
| Gene (*Oryzias latipes*) | *ptpn11* | Ensemble genome browser | Ensemble (release 103): ENSORLG00000000470.2 CDS | Medaka *ptpn11a* - protein tyrosine phosphatase non-receptor type 11 |
| Gene (*Homo sapiens*) | *ptpn11* | Ensemble genome browser | Ensemble (release 103): ENSG00000179295.18 CDS | Human protein tyrosine phosphatase non-receptor type 11 |
| Commercial assay, kit | NEBuilder HiFi DNA Assembly Cloning kit | New England Biolabs | Catalog #E5520S | |
| Commercial assay, kit | mMessage mMachine Sp6 Transcription Kit | Thermo Fisher Scientific | Catalog #AM1340 | |
| Commercial assay, kit | mMessage mMachine T7 Transcription Kit | Thermo Fisher Scientific | Catalog #AM1344 | |
| Commercial assay, kit | T7 MEGAscript Kit | Thermo Fisher Scientific | Catalog #AM1334 | |
| Commercial assay, kit | InnuPREP Gel Extraction Kit | Analytik Jena | Catalog #845-KS-5030250 | |
| Commercial assay, kit | Monarch DNA Gel Extraction Kit | New England Biolabs | Catalog #T1020 | |
| Commercial assay, kit | RNeasy Mini Kit | Qiagen | Catalog #74,106 | |
| Peptide, recombinant protein | Q5 High-Fidelity DNA Polymerase | New England Biolabs | Catalog #M0491 | |
| Peptide, recombinant protein | Q5 Hot High-Fidelity DNA Polymerase | New England Biolabs | Catalog #M0493 | |
| Chemical compound, drug | 2,3-Butanedione 2-monoxime (BDM) | Abcam | Catalog #ab120616 | |
| Chemical compound, drug | N-Phenylthiourea (PTU) | Sigma-Aldrich | Catalog #P7629 | |
| Chemical compound, drug | Ethyl 3-aminobenzoate methanesulfonate salt (Tricaine) | Sigma-Aldrich | Catalog #A5040 | |
| Software, algorithm | ACEofBASEs | This paper | https://aceofbases.cos.uni-heidelberg.de | |

*Continued*

| Reagent type (species) or resource | Designation | Source or reference | Identifiers | Additional information |
|---|---|---|---|---|
| Software, algorithm | EditR | *Kluesner et al., 2018* | https://moriaritylab.shinyapps.io/editr_v10/ | |
| Software, algorithm | Geneious | Biomatters | Version 8.1.9 | |
| Software, algorithm | Fiji distribution of ImageJ | *Schindelin et al., 2012* | Version 2.0.0 | |
| Software, algorithm | Adobe Illustrator | Adobe | Version 23.2.1 | |
| Software, algorithm | R, R studio | *R Development Core Team, 2020* | https://www.R-project.org/ | |
| Software, algorithm | R package | *Wickham et al., 2019* | Tidyverse | |
| Software, algorithm | R package | *Wickham, 2016* | ggplot2 | |
| Software, algorithm | R package | *Kassambara, 2020* | ggpubr | |
| Software, algorithm | R package | *Wickham, 2011* | plyr | |
| Software, algorithm | R package | *Wickham et al., 2020* | dplyr | |

## Fish lines and husbandry

Medaka (*Oryzias latipes*) and zebrafish (*Danio rerio*) stocks were maintained (fish husbandry, permit number 35–9185.64/BH Wittbrodt) and experiments (permit numbers 35–9185.81 /G-145/15 and 35–9185.81/G-271/20 Wittbrodt) were performed in accordance with local animal welfare standards (Tierschutzgesetz §11, Abs. 1, Nr. 1) and with European Union animal welfare guidelines (*Bert et al., 2016*). Fish were maintained in closed stocks and constant recirculating systems at 28 °C on a 14 hr light/10 hr dark cycle. The fish facility is under the supervision of the local representative of the animal welfare agency. The following medaka lines: Cab as wild-type (*Loosli et al., 2000*), Cab (*myl7::EGFP*) (*Gierten et al., 2020*), HdrR (*myl7::EGFP myl7::H2A-mCherry*) (*Hammouda et al., 2021*). The zebrafish line AB was used in this study as wild-type.

## Plasmids

To generate the pCS2+_evoBE4max plasmid, pBT281(evoAPOBEC1-BE4max; Addgene plasmid #122611, a gift from David Liu; *Thuronyi et al., 2019*) was assembled into pCS2+ (*Rupp et al., 1994*) by NEBuilder HiFi DNA Assembly (NEB) using Q5 polymerase PCR products (NEB). pCMV_AncBE4max (Addgene plasmid #112094; *Koblan et al., 2018*) and pBABE8e (Addgene plasmid #138489; *Richter et al., 2020*) are gifts from David Liu. pGGEV_4_BE4-Gam was used as previously published (*Thumberger et al., 2022*).

Oligo sequences used to clone pCS2+(evoBE4max):

| Primer name | Primer sequence 5′–3′ |
|---|---|
| forward | |
| pCS2 +backbone | GCCTCTAGAACTATAGTGAGTCG |
| evoBE4max fragment 1 | GTTCTTTTTGCAGGATCCCATTTACCATGAAACGGACAGCCGAC |
| evoBE4max fragment 2 | CAAGGACAAGGACTTCCTG |
| reverse | |

*Continued on next page*

*Continued*

| Primer name | Primer sequence 5'–3' |
|---|---|
| pCS2 +backbone | ATGGGATCCTGCAAAAAGAACAAG |
| evoBE4max fragment 1 | CAGGAAGTCCTTGTCCTTG |
| evoBE4max fragment 2 | CTCACTATAGTTCTAGAGGCTTAGACTTTCCTCTTCTTCTTGG |

## ACEofBASEs (A Careful Evaluation of Base Edits)

ACEofBASEs is a web-tool that was built based on the architecture of CCTop (*Stemmer et al., 2015*) to identify target sequences in the query sequence and predict possible off-target sites. On top of that ACEofBASEs will first infer if the query sequence if spliced, that is it does not contain intronic sequences in between, by aligning it to the genome of the target species. The tool blat (*Kent, 2002*) is used for that purpose allowing a minimum identity of 98%. In the case that the query sequence is identified as spliced, the intronic part will be added to reconstruct the genomic sequence, otherwise the query sequence will be taken as is. After that, only target sites with editable bases in the editing window are kept and the corresponding translation is depicted, depending on the frame chosen with respect to the start of the query sequence, using the standard translation code.

## sgRNA design and synthesis

sgRNAs were designed with ACEofBASEs (https://aceofbases.cos.uni-heidelberg.de), applying the following parameters: presence of at least one A or C nucleotide in the respective base editing window of the respective base editor variant, limitation of maximal 2 mismatches in the core region (set to 12 nucleotides adjacent to PAM) or maximal 4 mismatches in the entire spacer sequence. All sgRNA target sites in the query sequence were evaluated for potential off-targets against the respective genome (Japanese medaka HdrR (*Oryzias latipes*) Ensembl V 103; Zebrafish (*Danio rerio*) Ensemble V 103) and sgRNAs were selected for the intended codon change. Oligos were designed and selected with substitution (i.e. replacement of the two most 5' nucleotides with Gs if necessary, to foster T7 in vitro transcription) for transcription from DR274 (DR274 was a gift from Keith Joung, Addgene plasmid #42250; *Hwang et al., 2013*).

List of sgRNAs used in this study each targeted locus.

| sgRNA description | Target site [PAM] 5'- > 3' | Reference |
|---|---|---|
| oca2-Q333 | GAAACCCAGGTGGCCATTGC[AGG] | *Lischik et al., 2019* |
| oca2-Q256 | GATCCAAGTGGAGCAGACTG[AGG] | *Lischik et al., 2019* |
| oca2-T306 | CACAATCCAGGCCTTCCTGC[AGG] | |
| dr_oca2_L293 | GTACAGCGACTGGTTAGTCA[TGG] | *Hammouda et al., 2019* |
| tnnt2a_Q114 | AGAGCGCCAAAAACGTCTTG[AGG] | *Meyer et al., 2020* |
| kcnh6a_Q11 | GGCGCTCCAGAACACCTATT[TGG] | |
| kcnh6a_T507 | TGAAGACAGCCCGACTGCTC[AGG] | |
| kcnh6a_R509 | GACAGCCCGACTGCTCAGGT[TGG] | |
| kcnh6a_L511/R512 | ACTGCTCAGGTTGGTGCGAG[TGG] | |
| kcnh6a_D521/R522 | CTGGACCGTTACTCGGAGTA[CGG] | |
| tnnt2c_R112 | AAGCACGAGTGGCTGAGGAG[AGG] | |
| s1pr2_R150 | AACATAGCGCTCTATAGCTA[TGG] | |
| s1pr2_R167 | TGCCGCATGTTTCTGCTGAT[AGG] | |
| GFP_C71 | AGCACTGCACGCCGTAGGTC[AGG] | *Hammouda et al., 2021* |

*Continued on next page*

*Continued*

| sgRNA description | Target site [PAM] 5'- > 3' | Reference |
|---|---|---|
| *dapk3_P204* | CGAGCCGCTCGGCCTGGAGG[CGG] | |
| *ube2b_R8* | TAGCCGTCTTCTTGCTGGTG[TGG] | |
| *usp44_E68* | TAACTTCCATAGCTAACGGG[TGG] | |
| *ptpn11_G504* | CCGCTCCAAGCGCTCGGGGA[TGG] | |
| *tnnt2a_e2-SA* | TTCAGATGAAGAGGGAG[AGG] | |
| *tnnt2a_D189_R190* | TCAAGATAGACTTAAGTAAG[TGG] | |
| *tnnt2a_W201* | CATCCACTCCCAAAGCTCCA[CGG] | |

## Base editing sgRNA rank score

To evaluate sgRNAs and rank them we used three characteristics, efficiency, dinucleotide preference, and editing window. The following describes how we, based on empirical data published previously and confirmed in this study, attributed scores.

In total we considered these, for three CBEs: BE4-Gam, ancBE4max and evoBE4max and ABE8e, the ABE we tested.

Our data suggests that BE4-Gam demonstrates efficiencies much lower than the three second generation editors tested, ancBE4max, evoBE4max and ABE8e. We therefore ascribed 5 points (BE4-Gam) and 10 points (second gen. editors) – the starting value points, when selecting a specific editor in the drop-down menu. To include dinucleotide preferences in our score, we decided to apply penalties for such dinucleotide contexts that are disfavoured when used with the selected editor. Moreover, we considered base editing windows. We considered two types of windows, such which observed editing (usually broader) and those with optimal efficiency.

Summary of penalty scores is provided:

| Editor | Efficiency (E) | Dinucleotide context penalty, DP (penalty factor, PF) | | | | Editing window on protospacer: peak (observed) activity | Editing window penalty (for outside of peak window editing) WP (PF = 0.8) |
|---|---|---|---|---|---|---|---|
| | | **T** | **C** | **A** | **G** | | |
| BE4-Gam | 5 | 0 (0) | 2 (0.4) | 3 (0.6) | 4 (0.8) | 3–10 (4-8) | 4 |
| ancBE4max | 10 | 0 (0) | 2 (0.2) | 2 (0.2) | 8 (0.8) | 3–9 (4-7) | 8 |
| evoBE4max | 10 | 0 (0) | 2 (0.2) | 8 (0.8) | 2 (0.2) | 1–11 (4-8) | 8 |
| ABE8e | 10 | 0 (0) | 0 (0) | 0 (0) | 0 (0) | 3–11 (4-8) | 8 |

We identified the cytosine along the protospacer that received the highest score value and ranked this sgRNA accordingly (*Figure 1—figure supplement 2*).

$$S_{efficiency} = E - DP - WP$$

E = efficiency.
DP = dinucleotide context penalty = E x penalty factor (PF).
WP = Editing window penalty = E x penalty factor (PF).
S = score.

## In vitro transcription of mRNA

pGGEV_4_BE4-Gam was linearised with SpeI and pCS2+_evoBE4max was linearised with NotI, then mRNA was transcribed in vitro for both with the mMessage mMachine Sp6 Transcription Kit. pCMV_AncBE4max and pBABE8e were both linearized with SapI and mRNA was transcribed in vitro with mMessage mMachine T7 Transcription Kit.

## Microinjection for F0 experiments

Medaka one-cell stage embryos were injected into the cytoplasm as previously described (*Rembold et al., 2006*). Zebrafish one-cell stage embryos were injected into the yolk. For medaka, injection

solutions contained 150 ng/µl base editor mRNA, 30 ng/µl of the respective sgRNAs and 20 ng/µl GFP mRNA as injection tracer. Control siblings were injected with 20 ng/µl GFP mRNA and 30 ng/µl sgRNA only or rather 20 ng/µl GFP mRNA and 150 ng/µl base editor mRNA only. For zebrafish injections, injection mixes contained 360 pg ABE8e mRNA, 40 pg *oca2-L293* sgRNA and 12 pg GFP mRNA as injection tracer. Injected embryos were incubated at 28 °C in zebrafish medium (*Westerfield, 2000*) or medaka embryo rearing medium (ERM) (*Becker et al., 2021*) and selected for GFP expression 7 hours post injection.

### Microinjection for F1 experiments (*Tnnt2a, Dapk3,* and *Usp44*)

To investigate the phenotype-genotype correlation of the intended base edit in edited embryos after germline transmission, microinjections with lower base editor mRNA concentrations ranging from 5 to 50 ng/µl were carried out to overcome the increased lethality rates. Surviving F0 larvae were raised to adulthood. Founder fish were identified by outcrossing F0 editants to wildtype fish, followed by genotyping of their offspring. For F1 phenotyping and genotyping founder mating pairs were set up.

### Genotyping

Single or pools of five randomly selected embryos from respective injection experiments and adult fin clips were lysed in DNA extraction buffer (0.4 M Tris/HCl pH 8.0, 0.15 M NaCl, 0.1% SDS, 5 mM EDTA pH 8.0, 1 mg/ml proteinase K) at 60 °C overnight. Proteinase K was inactivated at 95 °C for 20 min and the solution was diluted 1:2 with nuclease-free water. Genomic DNA was precipitated in 300 mM sodium acetate and 3 x vol. absolute ethanol at 20,000 x g at 4 °C. Precipitated genomic DNA was resuspended in TE buffer (10 mM Tris pH 8.0, 1 mM EDTA in RNAse-free water) and stored at 4 °C. Genotyping-PCR was performed in 1 x Q5 reaction buffer, 200 µM dNTPs, 200 µM forward and reverse primer, 1 µl precipitated DNA sample and 0.3 U Q5 polymerase (NEB).

Oligo sequences used to amplify the target loci:

| oligo pair description | Fwd (5'–3') | Rev (5'–3') | Reference |
| --- | --- | --- | --- |
| oca2_T306&Q33_seq | GTTAAAACAGTTTCTTAAAAAGAACAGGA | AGCAGAAGAAATGACTCAACATTTTG | *Hammouda et al., 2019* |
| oca2_CBE_ABE_OT1_seq | CTCTGGTTACACAATGCGCG | ACAGTAGCATGCAGGCTCTC | |
| oca2_CBE_ABE_OT2_seq | TTGTGATGCTGCTGTTGCAC | CCCTTAATGGACGAGCAGCA | |
| oca2_CBE_OT3_seq | AGTGTCTGGATTGGATCAGTAGATG | GTGCCTGACCACTCTGACAT | |
| oca2_ABE_OT3_seq | AGAGTGGGACTTTAAAGATGCACA | ACTTGTGCAGCACTTTGGATG | |
| dr_oca2_L293_seq | ACAGGTGCTGTATAATTGGACCAT | AAAGAGTGGTCATAAACGGCTACT | *Hammouda et al., 2019* |
| tnnt2a_Q114_seq | TGGAGAAAGACCTGATGGAGC | TTCCCGCTCCTCTTCTCTGT | |
| kcnh6a_Q11_seq | ACATCCTGCATCTGCCATCG | GCAGGTGCAGTGAACCAAAA | |
| kcnh6a_S4domain_seq | GCTTTGCAAGGTATAGAGCACAG | AACGTTGCCAAAACCCACAC | |
| tnnt2c_R112_seq | GTGCCTAACATGGTCCCTCC | ACCTCTGGTGGTCACTGACT | |
| s1pr2_R150_seq | CCTGGTTCTGATGGCTGTGT | CCCAGCACTATTGTGACCGT | |
| GFP_C71_seq | GTGAGCAAGGGCGAGGAGCT | CTTGTACAGCTCGTCCATGC | *Gutierrez-Triana et al., 2018* |
| dapk3_P204_seq | CCTTAAGGAGGCAGCGAGTC | ACAGACATGAGTGTGGGCTG | |
| ube2b_R8_seq | AGGCGTTTTAATTGACATTTTGACG | CCTGTCTGGCTTCATAGACTGT | |

*Continued on next page*

*Continued*

| oligo pair description | Fwd (5'–3') | Rev (5'–3') | Reference |
|---|---|---|---|
| USP44_E68_seq | TCTAGCTTTTT GGCTCCCCG | CTGTAGCTCC TGTGCTCCAC | |
| ptpn11_G504_seq | ACCTTTTCTCTG AACTGTCGTGT | AGGTCGGAC AGCGAGTACT | |
| tnnt2a_ex2-SA_seq | TGAAGGAGGAA TGCATCTCTGAC | TGTAAATAGCCAA GCTAATGGAAGC | |
| tnnt2a_D189_R190_seq | CCAGTCTTCAC TTTGGAGGCT | TTCATGATATTG TTTAACTCAAA GGACAGA | |
| tnnt2a_W201_seq | AGCATCAGCAG AAGAGTTCCG | GTTAGTGAAGAA CTTGGGTGACG | |

The conditions were: 98 °C 2 min, 30 cycles of 98 °C 30 s, annealing for 20 s and 72 °C 30 s per kb, and a final extension time of 5 min at 72 °C. Following agarose gel electrophoresis, amplicons were gel purified with the InnuPREP Gel Extraction Kit (Analytik Jena) or the Monarch DNA Gel Extraction Kit (NEB), and submitted for Sanger sequencing (Eurofins Genomics). Sequence analysis was performed in Geneious (R8) and the transition rates of base editing experiments were estimated using EditR (*Kluesner et al., 2018*).

## Targeted amplicon sequencing

Samples used in this study analyzed by targeted amplicon sequencing:

| Sample | N embryos in sample | Replicates | Reads aligned |
|---|---|---|---|
| oca2-Q333 control | pool of 10 | 1 | 291,384 |
| oca2-Q333 BE4-Gam | pool of 10 | 1 | 293,231 |
| oca2-Q333 ancBE4max | pool of 10 | 1 | 291,908 |
| oca2-Q333 evoBE4max | pool of 10 | 1 | 351,843 |
| ABE8e control | pool of 5 | 1 | 24,470 |
| oca2-Q333 ABE8e | pool of 5 | 3 | 22,343–57,984 |
| ABE8e control | 1 | 1 | 14,653 |
| GFP-C71 ABE8e | 1 | 4 | 10,696–66,311 |
| tnnt2a-Q114 control | 1 | 1 | 7,094 |
| tnnt2a-Q114 evoBE4max | 1 | 5 | 2,481–48,791 |
| ABE8e control | 1 | 1 | 11,387 |
| kcnh6a-R512 ABE8e | 1 | 4 | 4,038–86,327 |
| oca2-Q333 evoBE4max (control) | 1 | 1 | 21,768 |
| ube2b-R8 evoBE4max (control) | 1 | 3 | 24,613–57,332 |

Samples were prepared by PCR amplifying the regions of interest (327–362 bp) using locus-specific primers with 5' partial Illumina adapter sequences from the same source of gDNA using Q5 Hot Start High-Fidelity DNA Polymerase (NEB). PCR products were run on a 1% agarose gel, specific bands were excised and cleaned up using the Monarch DNA Gel Extraction Kit (NEB). Samples were submitted either directly (oca2 control, BE4-Gam, ancBE4max and evoBE4max) at 25 ng/µl or one PCR product, for each of the five different loci was pooled to equilmolarity at 25 ng/µl and submitted to GeneWiz (Azenta Life Sciences) for sequencing (Amplicon-EZ: Illumina MiSeq, 2 × 250 bp sequencing, paired-end).

Oligo sequences containing Illumina adapter sequences used to amplify the target loci:

| Oligo description | Oligo sequence (5'- > 3'), adapter sequence |
| --- | --- |
| oca2_Q333_HTS_F | ACACTCTTTCCCTACACGACGCTCTTCCGATCT CGTTAGAGTGGTATGGAGAACTGT |
| oca2_Q333_HTS_R | GACTGGAGTTCAGACGTGTGCTCTTCCGATCT ATGGTCCTCACATCAGCAGC |
| GFP_C71_HTS_genewiz_F | ACACTCTTTCCCTACACGACGCTCTTCCGATC TTCCGATCTCGTAAACGGCCACAAGTTCAG |
| GFP_C71_HTS_genewiz_R | GACTGGAGTTCAGACGTGTGCTCTTCCGATC TTTGCCGTCCTCCTTGAAGTC |
| kcnh6a_S4domain_HTS_genewiz_F | ACACTCTTTCCCTACACGACGCTCTTCCGAT CTAGTTTGCTGTGTACCTCCAGTT |
| kcnh6a_S4domain_HTS_genewiz_R | GACTGGAGTTCAGACGTGTGCTCTTCCGAT CTATCTTCATACCGCCCACACG |
| tnnt2a_Q114_HTS_genewiz_F | ACACTCTTTCCCTACACGACGCTCTTCCGAT CTTGAGAGCAGAAAGAAAGAGGAGG |
| tnnt2a_Q114_HTS_genewiz_R | GACTGGAGTTCAGACGTGTGCTCTTCCGAT CTTTGCGTCATCCTCTGCTCTC |
| ube2b_R8Q_HTS_genewiz_F | ACACTCTTTCCCTACACGACGCTCTTCCGAT CTAGCGAACTCCGTCACCTTAAAT |
| ube2b_R8Q_HTS_genewiz_R | GACTGGAGTTCAGACGTGTGCTCTTCCGAT CTCCTGTCTGGCTTCATAGACTGT |

## Analysis and plotting of next-generation sequencing data

Amplicon sequencing data were analyzed with CRISPResso2 v.2.1.2 using the default –base_editor_ output parameters (w 10, wc –10) (*Clement et al., 2019*). Demultiplexing was achieved by mapping to the five different loci, respectively.

Downstream analysis was conducted using R v.3.6.3 in R studio (packages: tidyverse, ggplot, ggpubr), with data for modified alleles and INDELs sourced from the 'CRISPResso_quantification_ of_editing_frequency.txt' output table. For each experiment, indels were quantified across the entire sequence calculating INDELs as follows: INDELs = "Only Insertions" + "Only Deletions" + "Insertions + Substitutions" + "Deletions + Substitutions" + "Insertions + Deletions + Substitutions". %INDELs were determined by calculating INDELs/reads aligned; %substitutions were determined by calculating "only substitutions"/reads aligned.

To obtain target nucleotide conversion rates and for the base composition plots around the proto-spacer region +/- 5 bp the "Nucleotide_percentage_table.txt" output table was used.

Sample+"Alleles_frequency_table_around" + locus.pdf files were color matched in Adobe Illustrator.

## Imaging

Gross morphology and heart dynamics of embryos were assessed with a Nikon SMZ18 Stereomicroscope equipped with a Nikon DS-Ri1 camera. For in vivo imaging of hearts medaka embryos were kept in 5 x PTU in 1 x ERM from 5 to 7 dpf to block thoracic and abdominal pigmentation. To capture morphological phenotypes, embryos were anesthetized in 1 x Tricaine diluted in 1 x ERM with subsequent induction of cardiac arrest by treatment with 50 mM BDM 40–60 min. For confocal imaging embryos were mounted laterally or ventrally (indicated in respective Figure legends) on glass-bottomed Petri dishes (MatTek Corporation, Ashland, MA) in 1% low melting agarose supplemented with 30 mM BDM and 1 x Tricaine. Hearts of live embryos were imaged with a Sp8 confocal microscope (Leica) with 10 x (air) or 20 x (glycerol) objectives.

## Analysis of SNVs in ClinVar

The ClinVar vcf file was downloaded from https://ftp.ncbi.nlm.nih.gov/pub/clinvar/vcf_GRCh38/ (date 2021-04-04). From this file, only the variants annotated as pathogenic (CLNSIG = Pathogenic) and single nucleotide (CLNVC = single_nucleotide_variant) were considered. Furthermore, as SNVs affecting the coding sequence were counted the variants annotated with any of these Sequence Ontology IDs: SO:0001578, SO:0001582, SO:0001583, SO:0001587. To identify which SNVs can be

modeled in medaka or zebrafish, the REST API from Ensembl (version 103, *Howe et al., 2021*) to retrieve the orthologous position defined by LastZ alignments was used. Only SNV whose coordinates were aligned uniquely and in which the aligned base is the identical in medaka or zebrafish to human were kept. To obtain the molecular consequence of the inferred SNVs in fish, Variant Effect Predictor from the Ensembl web site was used (*McLaren et al., 2016*).

## Analysis and data visualization

Images were processed with the Fiji distribution of ImageJ (*Schindelin et al., 2012*). Analysis and graphical data visualization were performed in R with the Tidyverse, ggplot2, ggpubr, plyr, and dplyr packages (*Kassambara, 2020*; *R Development Core Team, 2020*; *Wickham, 2016*; *Wickham, 2011*; *Wickham et al., 2020*; *Wickham et al., 2019*). Figures were assembled in Adobe Illustrator. Sample size (**n**) and number of independent experiments are mentioned in every figure/figure legend or the main text. No statistical methods were used to predetermine sample sizes. The experimental groups were allocated randomly, and no blinding was done during allocation.

## Acknowledgements

This research was supported by the German Science funding agency (DFG, FOR2509 project 10 (WI 1824/9-1) and EXC 2082/1 (3DMM2O) Wittbrodt C3), the European Research Council Synergy Grant IndiGene (number 810172) to JW and the DZHK (German Centre for Cardiovascular Research, number 81X2500189) to JW and JG. JG was supported by the Deutsche Herzstiftung eV (S/02/17) and by an Add-On Fellowship for Interdisciplinary Science of the Joachim Herz Stiftung. AC, BW and JG are members/alumni of the Heidelberg Biosciences International Graduate School (HBIGS). We thank S Lemke, O Hammouda, T Tavhelidse, L Doering, F Farkas and all members of the Wittbrodt lab for their critical, constructive feedback on the manuscript. We thank T Kellner and R Müller for excellent technical support and M Majewsky, E Leist, S Erny and A Saraceno for expert fish husbandry.

## Additional information

### Funding

| Funder | Grant reference number | Author |
|--------|------------------------|--------|
| Deutsche Forschungsgemeinschaft | WI 1824/9-1 | Joachim Wittbrodt |
| H2020 European Research Council | 810172 | Joachim Wittbrodt |
| Deutsches Zentrum für Herz-Kreislaufforschung | 81X2500189 | Joachim Wittbrodt Jakob Gierten |
| Deutsche Herzstiftung | S/02/17 | Jakob Gierten |
| Joachim Herz Stiftung | Add-on Fellowship | Jakob Gierten |
| Deutsche Forschungsgemeinschaft | 3DMM2O, EXC 2082/1 Wittbrodt C3 | Joachim Wittbrodt |

The funders had no role in study design, data collection and interpretation, or the decision to submit the work for publication.

### Author contributions

Alex Cornean, Conceptualization, Data curation, Formal analysis, Investigation, Methodology, Validation, Visualization, Writing – original draft, Writing – review and editing; Jakob Gierten, Conceptualization, Investigation, Methodology, Writing – original draft, Writing – review and editing; Bettina Welz, Data curation, Formal analysis, Investigation, Methodology, Validation, Writing – review and editing; Juan Luis Mateo, Conceptualization, Formal analysis, Methodology, Software, Writing – review and editing; Thomas Thumberger, Conceptualization, Methodology, Software, Writing – review

and editing; Joachim Wittbrodt, Conceptualization, Funding acquisition, Project administration, Resources, Supervision, Writing – original draft, Writing – review and editing

### Author ORCIDs
Alex Cornean http://orcid.org/0000-0003-3727-7057
Jakob Gierten http://orcid.org/0000-0001-8143-1918
Juan Luis Mateo http://orcid.org/0000-0001-9902-6048
Thomas Thumberger http://orcid.org/0000-0001-8485-457X
Joachim Wittbrodt http://orcid.org/0000-0001-8550-7377

### Decision letter and Author response
Decision letter https://doi.org/10.7554/eLife.72124.sa1
Author response https://doi.org/10.7554/eLife.72124.sa2

---

## Additional files

### Supplementary files
• Supplementary file 1. Cytosine base editing efficiencies. Shows nucleotide position (of CDS) and corresponding amino acid with changes. Note: only cytosines on the protospacer with clear editing are shown. *sgRNA on complementary strand

• Supplementary file 2. Adenine base editing efficiencies for ABE8e. Shows nucleotide position (of CDS) and corresponding amino acid with changes. Note: only adenines on the protospacer with clear editing are shown. *sgRNA on complementary strand #averaged over single *oca2-Q333* and pooled injections

• Supplementary file 3. Cytosine base editing efficiencies at GWAS validation genes. Shows nucleotide position (of CDS) and corresponding amino acid with changes. Note: only cytosines on the protospacer with clear editing are shown. *sgRNA on complementary strand

• Supplementary file 4. Editing efficiency estimation by Sanger vs Illumina sequencing. Comparison of target nucleotide editing efficiency and overview of indel frequency from Illumina data.

• Transparent reporting form

• Source code 1. ACEofBASEs source code.

### Data availability
Source code for ACEofBASEs has been provided.

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
