## [Editor Report]

This is an outstanding new method using base editor technology for introducing precise mutations in zebrafish and medaka vertebrate model systems. The approach is some of the strongest evidence to date that F0 functional analyses are becoming practical for screening work, with germline confirmation now rapidly possible as well due to the precise nature of the mutagenesis tool.

---

## [Decision Letter]

**Decision letter after peer review:**

Thank you for submitting your article "Precise in vivo functional analysis of DNA variants with base editing using ACEofBASEs target prediction" for consideration by *eLife*. Your article has been reviewed by 3 peer reviewers, one of whom is a member of our Board of Reviewing Editors, and the evaluation has been overseen by Richard White as the Senior Editor. The following individual involved in review of your submission has agreed to reveal their identity: Jeffrey J Essner (Reviewer #3).

Your paper is considered potentially of high interest to the field. However, there are several noted questions/concerns that are essential to be addressed:

1. F0 work – there are several key issues, including the substantive delay noted in the tnnt2a F0 embryos not noted in the ENU or insertional mutant alleles. Reconciling whether this is due to the specific allele they are making versus unanticipated negative consequences in the injected animals needs to be resolved.

2. The presentation suggests that base editors will not result in double-stranded DNA breaks. That is not the case in other systems due to the integral single-stranded DNA 'nicking' in base editors. In addition, the rapidly dividing zebrafish embryo will convert a nick in DNA into a full double-stranded break upon replication. Either the language included in the paper needs to be substantively revised, or experimental data that measures this issue needs to be documented.

Germline work

3. The authors present 8 different loci with good F0 success. However, there have been false-positives from F0 gene editing science in the field. In addition, the noted differences in the phenotype from tnnt2a and prior alleles is also suggesting some likely limitations on F0 work. The authors need to provide key comparison data from their gene editing via germline propagation for at least tnnt2a and one or two of the novel cardiac mutants that show modest penetrance in the F0 screening work.

*Reviewer #2 (Recommendations for the authors):*

The manuscript main strength relies on the bioinformatic tool presented as well as on the various base editor tools tested hand by hand both in medaka and zebrafish.

I'm less impressed by the analysis of F0 "editants" regarding the cardiac phenotype as from the data showed here it doesn't seem that this will be a possible approach especially when the putative phenotypes are not predicted. Without a stronger validation of the observed phenotypes, I would predict that the community would be discouraged to use the F0 analysis approach.

In particular I think the following points should be addressed:

1) Few times in the manuscript the authors claim that the base editor act specifically and "no other DNA sequence changes such as indels or unwanted editing were observed around the locus" (lines 196-197 and again later in the text). This point should be supported by NGS of the locus as Sanger sequence analysis cannot detect mistakes below 5% in the best conditions.

2) The tnnt2a editants show a strong developmental delay. This is particularly worrisome as the corresponding zebrafish mutants were morphologically normal. Is this a side effect of the injections? Furthermore, the ancBE4max injected embryos show no silent heart phenotypes. Based on these results the involvement of tnnt2a in heart contractility would not be obvious at all if not for the previously known sih mutants. Have the authors isolated stable mutants for this line? How the editants compare morphologically to the Madaka morphants or mutants? And how editants in zebrafish compare to the well characterized sih mutant line? As it is the editants seem to suffer of severe unspecific injection effects and they seem of little use.

3) A similar problem is presented in the data shown in figure 6 and Fig6sup3. Here the "Cardiac" phenotype is quite small, sometimes smaller than the fraction of dead embryos and only two times what is observed in the oca2 control editants (that presumably represent the background noise following base editor injections). Are these results really significant? they should be compared to stable lines as they could well represent injection "cardiac defects" artifacts that are notoriously common following morpholino injections, for instance.

*Reviewer #3 (Recommendations for the authors):*

Given that injection itself can produce phenotypes, it would have been good to see these alleles passed through the germline and generate phenotypes in subsequent generations. This would not only show that the alleles can be recovered, but also that the phenotypes observed in the F0 generation are not a result of injection. However, other studies with base editors have shown there is a strong correlation between somatic mutagenesis and germline transmission. The oca2 mutations somewhat speak to the specificity of the mutagenesis, but phenotypes other than defects in pigmentation were also observed (dead and abnormal). This should not hold up publication, but if some of this data is available, it would be good to include.

---

## [Author Response]

Your paper is considered potentially of high interest to the field. However, there are several noted questions/concerns that are essential to be addressed:1. F0 work – there are several key issues, including the substantive delay noted in the tnnt2a F0 embryos not noted in the ENU or insertional mutant alleles. Reconciling whether this is due to the specific allele they are making versus unanticipated negative consequences in the injected animals needs to be resolved.

We thank the referees and editors for pointing out the discrepancy between previously reported zebrafish mutant alleles with particular loss of cardiac contraction (silent heart) and the observed medaka “silent heart” phenotype, associated with a global developmental delay.

We have performed two sets of additional experiments to address this issue, validating the medaka tnnt2a silent-heart phenotype, first targeting tnnt2a at another site to generate additional alleles and second to establish stable homozygous mutants in F1.

1) Creating a different PTC (W201X) using the tnnt2a-W201 sgRNA with evoBE4max microinjections. With this independent sgRNAtargeting a different site of the protein, we obtained robust (91%) medaka silent-heart phenotypes fully resembling the initially observed phenotypes when introducing a PTC at position Q114 (Q114X) (Figure 5-supplement 2).

2) To validate the relevance of the F0 editant phenotype, we established, raised and in-crossed heterozygous F0 founder fish with germline transmission of the Q114X allele. The resulting F1 homozygous tnnt2a-Q114X mutant embryos displayed the same medaka silent heart phenotype initially observed (Figure 5f’) and were undistinguishable from the F0 editants.

While these results confirm our initial tnnt2a editant observations, they also highlight a distinct, broader function of tnnt2a in medaka compared to zebrafish, where the mutant phenotype is limited to the heart.

These findings are fully detailed in the revised version of the manuscript.

2. The presentation suggests that base editors will not result in double-stranded DNA breaks. That is not the case in other systems due to the integral single-stranded DNA 'nicking' in base editors. In addition, the rapidly dividing zebrafish embryo will convert a nick in DNA into a full double-stranded break upon replication. Either the language included in the paper needs to be substantively revised, or experimental data that measures this issue needs to be documented.

We thank the referees for raising this point. We have extended our analysis and have included Amplicon-seq to reveal and quantify alleles not immediately uncovered in the Sanger analysis. This NGS analysis revealed a potential nick-dependent indel formation in editants of 4.9% and 15.8%.

We analyzed all editors tested at the medaka oca2-Q333 locus and incorporated our findings in the new Figure 3, connected to Figure 2, as the samples for the NGS analysis were the same as used for Sanger sequencing.

Additionally, we performed NGS analysis for ABE8e GFP-C71 experiments (Figure 4g), evoBE4max tnnt2a-Q114 experiments (Figure 5e), ABE8e kcnh6a-R512 experiments (Figure 6f) and evoBE4max ube2b-R8 experiments (Figure 7-supplement 3).

We summarized our findings and directly compared the target C or A conversion efficiencies obtained by Sanger sequencing as opposed to the newly acquired Illumina data on the same samples in Table 5. Moreover, we have removed results obtained by TIDE analysis to avoid confusion from the main text and Tables 1-3, as well as the corresponding Materials and Method section.

We have taken care to quantify editing and indel formation in response to the editing attempts in the revised version of the manuscript.

Germline work3. The authors present 8 different loci with good F0 success. However, there have been false-positives from F0 gene editing science in the field. In addition, the noted differences in the phenotype from tnnt2a and prior alleles is also suggesting some likely limitations on F0 work. The authors need to provide key comparison data from their gene editing via germline propagation for at least tnnt2a and one or two of the novel cardiac mutants that show modest penetrance in the F0 screening work.

We thank the referees for raising this point. We have extended our analysis and have included Amplicon-seq to reveal and quantify alleles not immediately uncovered in the Sanger analysis. This NGS analysis revealed a potential nick-dependent indel formation in editants of 4.9% and 15.8%.

We analyzed all editors tested at the medaka oca2-Q333 locus and incorporated our findings in the new Figure 3, connected to Figure 2, as the samples for the NGS analysis were the same as used for Sanger sequencing.

Additionally, we performed NGS analysis for ABE8e GFP-C71 experiments (Figure 4g), evoBE4max tnnt2a-Q114 experiments (Figure 5e), ABE8e kcnh6a-R512 experiments (Figure 6f) and evoBE4max ube2b-R8 experiments (Figure 7-supplement 3).

We summarized our findings and directly compared the target C or A conversion efficiencies obtained by Sanger sequencing as opposed to the newly acquired Illumina data on the same samples in Table 5. Moreover, we have removed results obtained by TIDE analysis to avoid confusion from the main text and Tables 1-3, as well as the corresponding Materials and Method section.

We have taken care to quantify editing and indel formation in response to the editing attempts in the revised version of the manuscript.